# EQUIFORMER: EQUIVARIANT GRAPH ATTENTION TRANSFORMER FOR 3D ATOMISTIC GRAPHS

**Yi-Lun Liao, Tess Smidt**
Massachusetts Institute of Technology
{ylliao, tsmidt}@mit.edu
https://github.com/atomicarchitects/equiformer

## ABSTRACT

Despite their widespread success in various domains, Transformer networks have yet to perform well across datasets in the domain of 3D atomistic graphs such as molecules even when 3D-related inductive biases like translational invariance and rotational equivariance are considered. In this paper, we demonstrate that Transformers can generalize well to 3D atomistic graphs and present Equiformer, a graph neural network leveraging the strength of Transformer architectures and incorporating $SE(3)/E(3)$-equivariant features based on irreducible representations (irreps). First, we propose a simple and effective architecture by only replacing original operations in Transformers with their equivariant counterparts and including tensor products. Using equivariant operations enables encoding equivariant information in channels of irreps features without complicating graph structures. With minimal modifications to Transformers, this architecture has already achieved strong empirical results. Second, we propose a novel attention mechanism called equivariant graph attention, which improves upon typical attention in Transformers through replacing dot product attention with multi-layer perceptron attention and including non-linear message passing. With these two innovations, Equiformer achieves competitive results to previous models on QM9, MD17 and OC20 datasets.

## 1 INTRODUCTION

Machine learned models can accelerate the prediction of quantum properties of atomistic systems like molecules by learning approximations of *ab initio* calculations  (Gilmer et al., 2017; Zhang et al., 2018b; Jia et al., 2020; Gasteiger et al., 2020a; Batzner et al., 2022; Lu et al., 2021; Unke et al., 2021; Sriram et al., 2022; Rackers et al., 2023). In particular, graph neural networks (GNNs) have gained increasing popularity due to their performance. By modeling atomistic systems as graphs, GNNs naturally treat the set-like nature of collections of atoms, encode the interaction between atoms in node features and update the features by passing messages between nodes. One factor contributing to the success of neural networks is the ability to incorporate inductive biases that exploit the symmetry of data. Take convolutional neural networks (CNNs) for 2D images as an example: Patterns in images should be recognized regardless of their positions, which motivates the inductive bias of translational equivariance. As for atomistic graphs, where each atom has its coordinate in 3D Euclidean space, we consider inductive biases related to 3D Euclidean group $E(3)$, which include equivariance to 3D translation, 3D rotation, and inversion. Concretely, some properties like energy of an atomistic system should be constant regardless of how we shift the system; others like force should be rotated accordingly if we rotate the system. To incorporate these inductive biases, equivariant and invariant neural networks have been proposed. The former leverages geometric tensors like vectors for equivariant node features (Thomas et al., 2018; Weiler et al., 2018; Kondor et al., 2018; Fuchs et al., 2020; Batzner et al., 2022; Brandstetter et al., 2022; Musaelian et al., 2022), and the latter augments graphs with invariant information such as distances and angles extracted from 3D graphs (Schütt et al., 2017; Gasteiger et al., 2020b;a; Liu et al., 2022; Klicpera et al., 2021).

A parallel line of research focuses on applying Transformer networks (Vaswani et al., 2017) to other domains like computer vision (Carion et al., 2020; Dosovitskiy et al., 2021; Touvron et al., 2020) and graph (Dwivedi & Bresson, 2020; Kreuzer et al., 2021; Ying et al., 2021; Shi et al., 2022) and has demonstrated widespread success. However, as Transformers were developed for sequence data (Devlin et al., 2019; Baevski et al., 2020; Brown et al., 2020), it is crucial to incorporate domain-related inductive biases. For example, Vision Transformer (Dosovitskiy et al., 2021) shows that adopting a pure Transformer to image classification cannot generalize well and achieves worse results than CNNs when trained on only ImageNet (Russakovsky et al., 2015) since it lacks inductive biases like translational invariance. Note that ImageNet contains over 1.28M images and the size

is already larger than that of many quantum properties prediction datasets (Ruddigkeit et al., 2012; Chmiela et al., 2017; Chanussot* et al., 2021). Therefore, this highlights the necessity of including correct inductive biases when applying Transformers to the domain of 3D atomistic graphs.

Despite their widespread success in various domains, Transformers have yet to perform well across datasets (Fuchs et al., 2020; Thölke & Fabritiis, 2022; Le et al., 2022) in the domain of 3D atomistic graphs even when relevant inductive biases are incorporated. In this work, we demonstrate that Transformers can generalize well to 3D atomistic graphs and present **Equiformer**, an equivariant graph neural network utilizing $SE(3)/E(3)$-**equi**variant features built from irreducible representations (irreps) and a novel attention mechanism to combine the 3D-related inductive bias with the strength of Trans**former**. First, we propose a simple and effective architecture, Equiformer with dot product attention and linear message passing, by only replacing original operations in Transformers with their equivariant counterparts and including tensor products. Using equivariant operations enables encoding equivariant information in channels of irreps features without complicating graph structures. With minimal modifications to Transformers, this architecture has already achieved strong empirical results (Index 3 in Table 6 and 7). Second, we propose a novel attention mechanism called equivariant graph attention, which improves upon typical attention in Transformers through replacing dot product attention with multi-layer perceptron attention and including non-linear message passing. Combining these two innovations, Equiformer (Index 1 in Table 6 and 7) achieves competitive results on QM9 (Ruddigkeit et al., 2012; Ramakrishnan et al., 2014), MD17 (Chmiela et al., 2017; Schütt et al., 2017; Chmiela et al., 2018) and OC20 (Chanussot* et al., 2021) datasets. For QM9 and MD17, Equiformer achieves overall better results across all tasks or all molecules compared to previous models like NequIP (Batzner et al., 2022) and TorchMD-NET (Thölke & Fabritiis, 2022). For OC20, when trained with IS2RE data and optionally IS2RS data, Equiformer improves upon state-of-the-art models such as SEGNN (Brandstetter et al., 2022) and Graphormer (Shi et al., 2022). Particularly, as of the submission of this work, Equiformer achieves the best IS2RE result when only IS2RE and IS2RS data are used and improves training time by $2.3\times$ to $15.5\times$ compared to previous models.

## 2 RELATED WORKS

We focus on equivariant neural networks here. We provide a detailed comparison between other equivariant Transformers and Equiformer and discuss other related works in Sec. B in appendix.

***SE(3)/E(3)-Equivariant GNNs.*** Equivariant neural networks (Thomas et al., 2018; Kondor et al., 2018; Weiler et al., 2018; Fuchs et al., 2020; Miller et al., 2020; Townshend et al., 2020; Batzner et al., 2022; Jing et al., 2021; Schütt et al., 2021; Satorras et al., 2021; Unke et al., 2021; Brandstetter et al., 2022; Thölke & Fabritiis, 2022; Le et al., 2022; Musaelian et al., 2022) operate on geometric tensors like type-$L$ vectors to achieve equivariance. The central idea is to use functions of geometry built from spherical harmonics and irreps features to achieve 3D rotational and translational equivariance as proposed in Tensor Field Network (TFN) (Thomas et al., 2018), which generalizes 2D counterparts (Worrall et al., 2016; Cohen & Welling, 2016; Cohen et al., 2018) to 3D Euclidean space (Thomas et al., 2018; Weiler et al., 2018; Kondor et al., 2018). Previous works differ in equivariant operations used in their networks. TFN (Thomas et al., 2018) and NequIP (Batzner et al., 2022) use graph convolution with linear messages, with the latter utilizing extra equivariant gate activations (Weiler et al., 2018). SEGNN (Brandstetter et al., 2022) introduces non-linear messages (Gilmer et al., 2017; Sanchez-Gonzalez et al., 2020) for irreps features, and the non-linear messages use the same gate activation and improve upon linear messages. SE(3)-Transformer (Fuchs et al., 2020) adopts an equivariant version of dot product (DP) attention (Vaswani et al., 2017) with linear messages, and the attention can support vectors of any type $L$. Subsequent works on equivariant Transformers (Thölke & Fabritiis, 2022; Le et al., 2022) follow the practice of DP attention and linear messages but use more specialized architectures considering only type-$0$ and type-$1$ vectors.

The proposed Equiformer incorporates all the advantages through combining MLP attention with non-linear messages and supporting vectors of any type. Compared to TFN, NequIP, SEGNN and SE(3)-Transformer, the proposed combination of MLP attention and non-linear messages is more expressive than pure linear or non-linear messages and pure MLP or dot product attention. Compared to other equivariant Transformers (Thölke & Fabritiis, 2022; Le et al., 2022), in addition to being more expressive, the proposed attention mechanism can support vectors of higher degrees (types) and involve higher order tensor product interactions, which can lead to better performance.

## 3 BACKGROUND

### 3.1 $E(3)$ EQUIVARIANCE

Atomistic systems are often described using coordinate systems. For 3D Euclidean space, we can freely choose coordinate systems and change between them via the symmetries of 3D space: 3D

translation, rotation and inversion ($\vec{r} \rightarrow -\vec{r}$). The groups of 3D translation, rotation and inversion form Euclidean group $E(3)$, with the first two forming $SE(3)$, the second being $SO(3)$, and the last two forming $O(3)$. The laws of physics are invariant to the choice of coordinate systems and therefore properties of atomistic systems are equivariant, e.g., when we rotate our coordinate system, quantities like energy remain the same while others like force rotate accordingly. Formally, a function $f$ mapping between vector spaces $X$ and $Y$ is equivariant to a group of transformation $G$ if for any input $x \in X$, output $y \in Y$ and group element $g \in G$, we have $f(D_X(g)x) = D_Y(g)f(x)$, where $D_X(g)$ and $D_Y(g)$ are transformation matrices parametrized by $g$ in $X$ and $Y$. For learning on 3D atomistic graphs, features and learnable functions should be $E(3)$-equivariant to geometric transformation acting on position $\vec{r}$. In this work, following previous works (Thomas et al., 2018; Kondor et al., 2018; Weiler et al., 2018) implemented in e3nn (Geiger et al., 2022), we achieve $SE(3)/E(3)$-equivariance by using equivariant features based on vector spaces of irreducible representations and equivariant operations for learnable functions. In the main text, we discuss $SE(3)$-equivariance and benchmark Equiformer with $SE(3)$-equivariance. We leave the discussion on inversion and $E(3)$-equivariance in Sec. A and present results of $E(3)$-equivariance in Sec. D.2 and Sec. F.4 in appendix.

## 3.2 IRREDUCIBLE REPRESENTATIONS

A group representation (Dresselhaus et al., 2007; Zee, 2016) defines transformation matrices $D_X(g)$ of group elements $g$ that act on a vector space $X$. For 3D Euclidean group $E(3)$, two examples of vector spaces with different transformation matrices are scalars and Euclidean vectors in $\mathbb{R}^3$, i.e., vectors change with rotation while scalars do not. To address translation symmetry, we operate on relative positions. The transformation matrices of rotation and inversion are separable and commute. We discuss irreducible representations of $SO(3)$ below and discuss inversion in Sec. A.3 in appendix.

Any group representation of $SO(3)$ on a given vector space can be decomposed into a concatenation of provably smallest transformation matrices called irreducible representations (irreps). Specifically, for group element $g \in SO(3)$, there are $(2L+1)$-by-$(2L+1)$ irreps matrices $D_L(g)$ called Wigner-D matrices acting on $(2L + 1)$-dimensional vector spaces, where degree $L$ is a non-negative integer. $L$ can be interpreted as an angular frequency and determines how quickly vectors change when rotating coordinate systems. $D_L(g)$ of different $L$ act on independent vector spaces. Vectors transformed by $D_L(g)$ are type-$L$ vectors, with scalars and Euclidean vectors being type-0 and type-1 vectors. It is common to index elements of type-$L$ vectors with an index $m$ called order, where $-L \leq m \leq L$.

**Irreps Features.** We concatenate multiple type-$L$ vectors to form $SE(3)$-equivariant irreps features. Concretely, irreps feature $f$ has $C_L$ type-$L$ vectors, where $0 \leq L \leq L_{max}$ and $C_L$ is the number of channels for type-$L$ vectors. We index irreps features $f$ by channel $c$, degree $L$, and order $m$ and denote as $f_{c,m}^{(L)}$. Different channels of type-$L$ vectors are parametrized by different weights but are transformed with the same $D_L(g)$. Regular scalar features correspond to only type-0 vectors.

**Spherical Harmonics.** Euclidean vectors $\vec{r}$ in $\mathbb{R}^3$ can be projected into type-$L$ vectors $f^{(L)}$ by using spherical harmonics (SH) $Y^{(L)}$: $f^{(L)} = Y^{(L)}(\frac{\vec{r}}{||\vec{r}||})$. SH are $E(3)$-equivariant with $D_L(g)f^{(L)} = Y^{(L)}(\frac{D_1(g)\vec{r}}{||D_1(g)\vec{r}||})$. SH of relative position $\vec{r}_{ij}$ generates the first set of irreps features. Equivariant information propagates to other irreps features through equivariant operations like tensor products.

## 3.3 TENSOR PRODUCT

Tensor products can interact different type-$L$ vectors. We discuss tensor products for $SO(3)$ below and those for $O(3)$ in Sec. A.4. The tensor product denoted as $\otimes$ uses Clebsch-Gordan coefficients to combine type-$L_1$ vector $f^{(L_1)}$ and type-$L_2$ vector $g^{(L_2)}$ and produces type-$L_3$ vector $h^{(L_3)}$:

$$h_{m_3}^{(L_3)} = (f^{(L_1)} \otimes g^{(L_2)})_{m_3} = \sum_{m_1=-L_1}^{L_1} \sum_{m_2=-L_2}^{L_2} C_{(L_1,m_1)(L_2,m_2)}^{(L_3,m_3)} f_{m_1}^{(L_1)} g_{m_2}^{(L_2)} \tag{1}$$

where $m_1$ denotes order and refers to the $m_1$-th element of $f^{(L_1)}$. Clebsch-Gordan coefficients $C_{(L_1,m_1)(L_2,m_2)}^{(L_3,m_3)}$ are non-zero only when $|L_1 - L_2| \leq L_3 \leq |L_1 + L_2|$ and thus restrict output vectors to be of certain types. For efficiency, we discard vectors with $L > L_{max}$, where $L_{max}$ is a hyper-parameter, to prevent vectors of increasingly higher dimensions.

We call each distinct non-trivial combination of $L_1 \otimes L_2 \rightarrow L_3$ a path. Each path is independently equivariant, and we can assign one learnable weight to each path in tensor products, which is similar to typical linear layers. We can generalize Eq. 1 to irreps features and include multiple channels

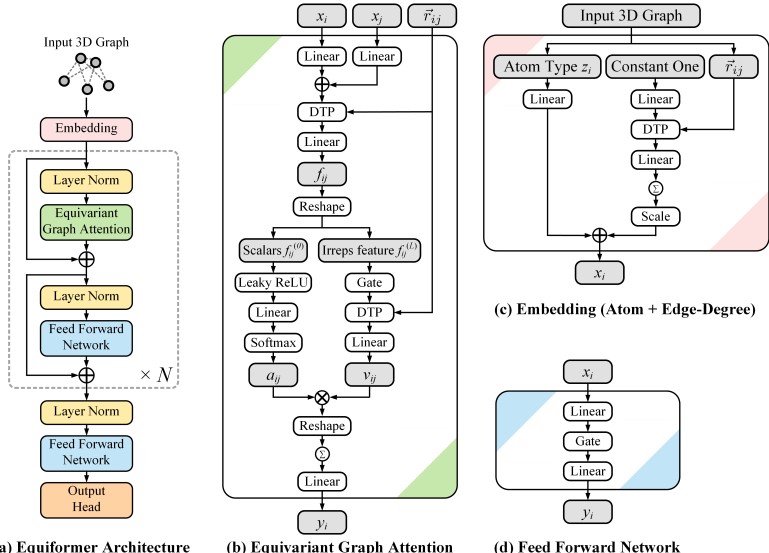

Figure 1: **Architecture of Equiformer.** We embed input 3D graphs with atom and edge-degree embeddings and process them with Transformer blocks, consisting of equivariant graph attention and feed forward networks. In this figure, "$\otimes$" denotes multiplication, "$\oplus$" denotes addition, and "DTP" stands for depth-wise tensor product. $\sum$ within a circle denotes summation over all neighbors. Gray cells indicate intermediate irreps features.

of vectors of different types through iterating over all paths associated with channels of vectors. In this way, weights are indexed by $(c_1, l_1, c_2, l_2, c_3, l_3)$, where $c_1$ is the $c_1$-th channel of type-$l_1$ vector in input irreps feature. We use $\otimes_w$ to represent tensor product with weights $w$. Weights can be conditioned on quantities like relative distances.

## 4    EQUIFORMER

First, we propose a simple and effective architecture, Equiformer with dot product attention and linear message passing, by only replacing original operations in Transformers with their equivariant counterparts and including tensor products for $SE(3)/E(3)$-equivariant irreps features. The equivariant operations are discussed in Sec. 4.1. The equivariant version of dot product attention can be found in Sec. C.3, and that of other modules in Transformers can be found in Sec. 4.3. Second, we propose a novel attention mechanism called equivariant graph attention in Sec. 4.2. The proposed Equiformer combines these two innovations and is illustrated in Fig. 1.

### 4.1    EQUIVARIANT OPERATIONS FOR IRREPS FEATURES

We discuss below equivariant operations, which serve as building blocks for equivariant graph attention and other modules, and analyze how they remain equivariant in Sec. C.1. They include the equivariant version of operations in Transformers and depth-wise tensor products as shown in Fig. 2.

**Linear.**    Linear layers are generalized to irreps features by transforming different type-$L$ vectors separately. Specifically, we apply separate linear operations to each group of type-$L$ vectors. We remove bias terms for non-scalar features with $L > 0$ as biases do not depend on inputs, and therefore, including biases for type-$L$ vectors with $L > 0$ can break equivariance.

**Layer Normalization.**    Transformers adopt layer normalization (LN) (Ba et al., 2016) to stabilize training. Given input $x \in \mathbb{R}^{N \times C}$, with $N$ being the number of nodes and $C$ the number of channels, LN calculates the linear transformation of normalized input as $\text{LN}(x) = \left( \frac{x - \mu_C}{\sigma_C} \right) \circ \gamma + \beta$, where $\mu_C, \sigma_C \in \mathbb{R}^{N \times 1}$ are mean and standard deviation of input $x$ along the channel dimension, $\gamma, \beta \in \mathbb{R}^{1 \times C}$ are learnable parameters, and $\circ$ denotes element-wise product. By viewing standard deviation as the root mean square value (RMS) of L2-norm of type-$L$ vectors, LN can be generalized to irreps features. Specifically, given input $x \in \mathbb{R}^{N \times C \times (2L+1)}$ of type-$L$ vectors, the output is $\text{LN}(x) = \left( \frac{x}{\text{RMS}_C(\text{norm}(x))} \right) \circ \gamma$, where $\text{norm}(x) \in \mathbb{R}^{N \times C \times 1}$ calculates the L2-norm of each type-$L$ vectors in $x$, and $\text{RMS}_C(\text{norm}(x)) \in \mathbb{R}^{N \times 1 \times 1}$ calculates the RMS of L2-norm with mean taken along the channel dimension. We remove means and biases for type-$L$ vectors with $L \neq 0$.

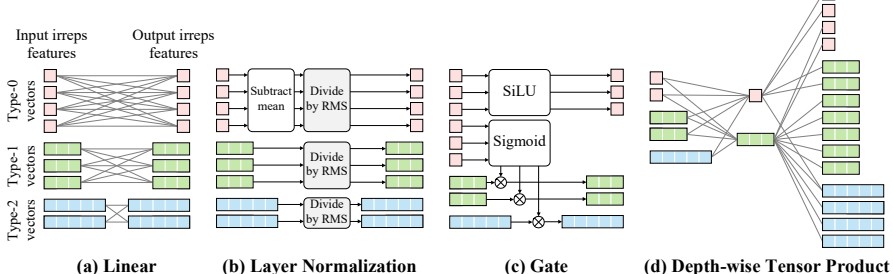

**(a) Linear**  **(b) Layer Normalization**  **(c) Gate**  **(d) Depth-wise Tensor Product**

Figure 2: **Equivariant operations used in Equiformer.** **(a)** Each gray line between input and output irreps features contains one learnable weight. **(b)** "RMS" denotes the root mean square value along the channel dimension. For simplicity, we have removed multiplying by $\gamma$ here. **(c)** Gate layers are equivariant activation functions where non-linearly transformed scalars are used to gate non-scalar irreps features. **(d)** The left two irreps features correspond to two input irreps features, and the rightmost one is the output irreps feature. The two gray lines connecting two vectors in the input irreps features and one vector in the output irreps feature form a path and contain one learnable weight. An alternative visualization of depth-wise tensor products can be found in Fig. 3 in appendix. We show $SE(3)$-equivariant operations here, which can be generalized to $E(3)$-equivariant features.

**Gate.** We use the gate activation (Weiler et al., 2018) for equivariant activation function as shown in Fig. 2(c). Typical activation functions are applied to type-0 vectors. For vectors of higher $L$, we multiply them with non-linearly transformed type-0 vectors for equivariance. Specifically, given input $x$ containing non-scalar $C_L$ type-$L$ vectors with $0 < L \leq L_{max}$ and $(C_0 + \sum_{L=1}^{L_{max}} C_L)$ type-0 vectors, we apply SiLU (Elfwing et al., 2017; Ramachandran et al., 2017) to the first $C_0$ type-0 vectors and sigmoid function to the other $\sum_{L=1}^{L_{max}} C_L$ type-0 vectors to obtain non-linear weights and multiply each type-$L$ vector with corresponding non-linear weights. After the gate activation, the number of channels for type-0 vectors is reduced to $C_0$.

**Depth-wise Tensor Product.** The tensor product defines interaction between vectors of different $L$. To improve its efficiency, we use the depth-wise tensor product (DTP), where one type-$L$ vector in output irreps features depends only on one type-$L'$ vector in input irreps features as illustrated in Fig. 2(d) and Fig. 3, with $L$ being equal to or different from $L'$. This is similar to depth-wise convolution (Howard et al., 2017), where one output channel depends on only one input channel. Weights $w$ in the DTP can be input-independent or conditioned on relative distances, and the DTP between two tensors $x$ and $y$ is denoted as $x \otimes_w^{DTP} y$. Note that the one-to-one dependence of channels can significantly reduce the number of weights and thus memory complexity when weights are conditioned on relative distances. In contrast, if one output channel depends on all input channels, in our case, this can lead to out-of-memory errors when weights are parametrized by relative distances.

## 4.2 EQUIVARIANT GRAPH ATTENTION

Self-attention (Vaswani et al., 2017; Veličković et al., 2018; Fuchs et al., 2020; Khan et al., 2021; Ying et al., 2021; Brody et al., 2022) transforms features sent from one spatial location to another with input-dependent weights. We use the notion from Transformers (Vaswani et al., 2017) and message passing networks (Gilmer et al., 2017) and define message $m_{ij}$ sent from node $j$ to node $i$ as follows:

$$m_{ij} = a_{ij} \times v_{ij} \tag{2}$$

where attention weights $a_{ij}$ depend on features on node $i$ and its neighbors $\mathcal{N}(i)$ and values $v_{ij}$ are transformed with input-independent weights. In Transformers and Graph Attention Networks (GAT) (Veličković et al., 2018; Brody et al., 2022), $v_{ij}$ depends only on node $j$. In message passing networks (Gilmer et al., 2017), $v_{ij}$ depends on features on nodes $i$ and $j$ with constant $a_{ij}$. The proposed equivariant graph attention adopts tensor products to incorporate content and geometric information and uses multi-layer perceptron attention for $a_{ij}$ and non-linear message passing for $v_{ij}$ as illustrated in Fig. 1(b).

**Incorporating Content and Geometric Information.** Given features $x_i$ and $x_j$ on target node $i$ and source node $j$, we combine the two features with two linear layers to obtain initial message $x_{ij} = \text{Linear}_{dst}(x_i) + \text{Linear}_{src}(x_j)$. $x_{ij}$ is passed to a DTP layer and a linear layer to consider

geometric information like relative position contained in different type-$L$ vectors in irreps features:

$$x'_{ij} = x_{ij} \otimes^{DTP}_{w(||\vec{r}_{ij}||)} \text{SH}(\vec{r}_{ij}) \quad \text{and} \quad f_{ij} = \text{Linear}(x'_{ij}) \tag{3}$$

where $x'_{ij}$ is the tensor product of $x_{ij}$ and spherical harmonics embeddings (SH) of relative position $\vec{r}_{ij}$, with weights parametrized by $||\vec{r}_{ij}||$. $f_{ij}$ considers semantic and geometric features on source and target nodes in a linear manner and is used to derive attention weights and non-linear messages.

**Multi-Layer Perceptron Attention.** Attention weights $a_{ij}$ capture how each node interacts with neighboring nodes. $a_{ij}$ are invariant to geometric transformation, and thus, we only use type-0 vectors (scalars) of message $f_{ij}$ denoted as $f_{ij}^{(0)}$ for attention. Note that $f_{ij}^{(0)}$ encodes directional information, as they are generated by tensor products of type-$L$ vectors with $L \geq 0$. Inspired by GATv2 (Brody et al., 2022), we adopts multi-layer perceptron attention (MLPA) instead of dot product attention (DPA) used in Transformers (Vaswani et al., 2017). In contrast to dot product, MLPs are universal approximators (Hornik et al., 1989; Hornik, 1991; Cybenko, 1989) and can theoretically capture any attention patterns. Given $f_{ij}^{(0)}$, we uses one leaky ReLU layer and one linear layer for $a_{ij}$:

$$z_{ij} = a^\top \text{LeakyReLU}(f_{ij}^{(0)}) \quad \text{and} \quad a_{ij} = \text{softmax}_j(z_{ij}) = \frac{\exp(z_{ij})}{\sum_{k \in \mathcal{N}(i)} \exp(z_{ik})} \tag{4}$$

where $a$ is a learnable vectors of the same dimension as $f_{ij}^{(0)}$ and $z_{ij}$ is a single scalar. The output of attention is the sum of value $v_{ij}$ multiplied by corresponding $a_{ij}$ over all neighboring nodes $j \in \mathcal{N}(i)$, where $v_{ij}$ can be obtained by linear or non-linear transformations of $f_{ij}$ as discussed below.

**Non-Linear Message Passing.** Values $v_{ij}$ are features sent from one node to another, transformed with input-independent weights. We first split $f_{ij}$ into $f_{ij}^{(L)}$ and $f_{ij}^{(0)}$, where the former consists of type-$L$ vectors with $0 \leq L \leq L_{max}$ and the latter consists of scalars only. Then, we perform non-linear transformation to $f_{ij}^{(L)}$ to obtain non-linear message:

$$\mu_{ij} = \text{Gate}(f_{ij}^{(L)}) \quad \text{and} \quad v_{ij} = \text{Linear}([\mu_{ij} \otimes^{DTP}_w \text{SH}(\vec{r}_{ij})]) \tag{5}$$

We apply gate activation to $f_{ij}^{(L)}$ to obtain $\mu_{ij}$. We use one DTP and a linear layer to enable interaction between non-linear type-$L$ vectors, which is similar to how we transform $x_{ij}$ into $f_{ij}$. Weights $w$ here are input-independent. We can also use $f_{ij}^{(L)}$ directly as $v_{ij}$ for linear messages.

**Multi-Head Attention.** Following Transformers (Vaswani et al., 2017), we can perform $h$ parallel equivariant graph attention functions given $f_{ij}$. The $h$ different outputs are concatenated and projected with a linear layer, resulting in the final output $y_i$ as illustrated in Fig. 1(b). Note that parallelizing attention functions and concatenating can be implemented with "Reshape".

## 4.3 Overall Architecture

For completeness, we discuss other modules in Equiformer here.

**Embedding.** This module consists of atom embedding and edge-degree embedding. For the former, we use a linear layer to transform one-hot encoding of atom species. For the latter, as depicted in the right branch in Fig. 1(c), we first transform a constant one vector into messages encoding local geometry with two linear layers and one intermediate DTP layer and then use sum aggregation to encode degree information (Xu et al., 2019; Shi et al., 2022). The DTP layer has the same form as that in Eq. 3. We scale the aggregated features by dividing with the squared root of average degrees in training sets so that standard deviation of aggregated features would be close to 1.

**Radial Basis and Radial Function.** Relative distances $||\vec{r}_{ij}||$ parametrize weights in some DTP layers. To reflect subtle changes in $||\vec{r}_{ij}||$, we represent distances with radial basis like Gaussian radial basis (Schütt et al., 2017) and radial Bessel basis (Gasteiger et al., 2020b;a). We transform radial basis with a learnable radial function to generate weights for those DTP layers. The function consists of a two-layer MLP, with each linear layer followed by LN and SiLU, and a final linear layer.

**Feed Forward Network.** Similar to Transformers, we use two equivariant linear layers and an intermediate gate activation for the feed forward networks in Equiformer.

**Output Head.** The last feed forward network transforms features on each node into a scalar. We perform sum aggregation over all nodes to predict scalar quantities like energy. Similar to edge-degree embedding, we divide the aggregated scalars with the squared root of average numbers of atoms.

| Methods | Task Units | $\alpha$ $a_0^3$ | $\Delta\varepsilon$ meV | $\varepsilon_{\text{HOMO}}$ meV | $\varepsilon_{\text{LUMO}}$ meV | $\mu$ D | $C_\nu$ cal/mol K | $G$ meV | $H$ meV | $R^2$ $a_0^2$ | $U$ meV | $U_0$ meV | ZPVE meV |
|---|---|---|---|---|---|---|---|---|---|---|---|---|---|
| NMP (Gilmer et al., 2017)† | | .092 | 69 | 43 | 38 | .030 | .040 | 19 | 17 | .180 | 20 | 20 | 1.50 |
| SchNet (Schütt et al., 2017) | | .235 | 63 | 41 | 34 | .033 | .033 | 14 | 14 | .073 | 19 | 14 | 1.70 |
| Cormorant (Anderson et al., 2019)† | | .085 | 61 | 34 | 38 | .038 | .026 | 20 | 21 | .961 | 21 | 22 | 2.03 |
| LieConv (Finzi et al., 2020)† | | .084 | 49 | 30 | 25 | .032 | .038 | 22 | 24 | .800 | 19 | 19 | 2.28 |
| DimeNet++ (Gasteiger et al., 2020a) | | **.044** | 33 | 25 | 20 | .030 | .023 | 8 | 7 | .331 | 6 | 6 | 1.21 |
| TFN (Thomas et al., 2018)† | | .223 | 58 | 40 | 38 | .064 | .101 | - | - | - | - | - | - |
| SE(3)-Transformer (Fuchs et al., 2020)† | | .142 | 53 | 35 | 33 | .051 | .054 | - | - | - | - | - | - |
| EGNN (Satorras et al., 2021)† | | .071 | 48 | 29 | 25 | .029 | .031 | 12 | 12 | .106 | 12 | 11 | 1.55 |
| PaiNN (Schütt et al., 2021) | | .045 | 46 | 28 | 20 | .012 | .024 | **7.35** | **5.98** | .066 | **5.83** | **5.85** | 1.28 |
| TorchMD-NET (Thölke & Fabritiis, 2022) | | .059 | 36 | 20 | 18 | **.011** | .026 | 7.62 | 6.16 | **.033** | 6.38 | 6.15 | 1.84 |
| SphereNet (Liu et al., 2022) | | .046 | 32 | 23 | 18 | .026 | **.021** | 8 | 6 | .292 | 7 | 6 | **1.12** |
| SEGNN (Brandstetter et al., 2022)† | | .060 | 42 | 24 | 21 | .023 | .031 | 15 | 16 | .660 | 13 | 15 | 1.62 |
| EQGAT (Le et al., 2022) | | .053 | 32 | 20 | 16 | **.011** | .024 | 23 | 24 | .382 | 25 | 25 | 2.00 |
| Equiformer | | .046 | **30** | **15** | **14** | **.011** | .023 | 7.63 | 6.63 | .251 | 6.74 | 6.59 | 1.26 |

Table 1: **MAE results on QM9 testing set.** † denotes using different data partitions.

| | Aspirin | | Benzene | | Ethanol | | Malonaldehyde | | Naphthalene | | Salicylic acid | | Toluene | | Uracil | |
|---|---|---|---|---|---|---|---|---|---|---|---|---|---|---|---|---|
| Methods | energy | forces | energy | forces | energy | forces | energy | forces | energy | forces | energy | forces | energy | forces | energy | forces |
| SchNet (Schütt et al., 2017) | 16.0 | 58.5 | 3.5 | 13.4 | 3.5 | 16.9 | 5.6 | 28.6 | 6.9 | 25.2 | 8.7 | 36.9 | 5.2 | 24.7 | 6.1 | 24.3 |
| DimeNet (Gasteiger et al., 2020b) | 8.8 | 21.6 | 3.4 | 8.1 | 2.8 | 10.0 | 4.5 | 16.6 | 5.3 | 9.3 | 5.8 | 16.2 | 4.4 | 9.4 | 5.0 | 13.1 |
| PaiNN (Schütt et al., 2021) | 6.9 | 14.7 | - | - | 2.7 | 9.7 | 3.9 | 13.8 | 5.0 | 3.3 | 4.9 | 8.5 | 4.1 | 4.1 | 4.5 | 6.0 |
| TorchMD-NET (Thölke & Fabritiis, 2022) | **5.3** | 11.0 | 2.5 | 8.5 | 2.3 | 4.7 | 3.3 | 7.3 | **3.7** | 2.6 | **4.0** | 5.6 | **3.2** | 2.9 | **4.1** | 4.1 |
| NequIP ($L_{max}=3$) (Batzner et al., 2022) | 5.7 | 8.0 | - | - | **2.2** | 3.1 | 3.3 | 5.6 | 4.9 | **1.7** | 4.6 | 3.9 | 4.0 | **2.0** | 4.5 | **3.3** |
| Equiformer ($L_{max}=2$) | **5.3** | 7.2 | **2.2** | **6.6** | **2.2** | 3.1 | 3.3 | 5.8 | **3.7** | 2.1 | 4.5 | 4.1 | 3.8 | 2.1 | 4.3 | **3.3** |
| Equiformer ($L_{max}=3$) | **5.3** | **6.6** | 2.5 | 8.1 | **2.2** | **2.9** | **3.2** | **5.4** | 4.4 | 2.0 | 4.3 | **3.9** | 3.7 | 2.1 | 4.3 | 3.4 |

Table 2: **MAE results on MD17 testing set.** Energy and force are in units of meV and meV/Å.

## 5 EXPERIMENT

We benchmark Equiformer on QM9 (Sec. 5.1), MD17 (Sec. 5.2) and OC20 (Sec. 5.3) datasets. Moreover, ablation studies (Sec. 5.4) are conducted to demonstrate that Equiformer with dot prodcut attention and linear message passing has already achieved strong empirical results on QM9 and OC20 datasets and verify that the proposed equivariant graph attention improves upon typical dot product attention in Transformer as well as dot product attention in other equivariant Transformers. Additional results of including inversion can be found in Sec. D.2 and Sec. F.4.

### 5.1 QM9

**Dataset.** The QM9 dataset (Ruddigkeit et al., 2012; Ramakrishnan et al., 2014) (CC BY-NC SA 4.0 license) consists of 134k small molecules, and the goal is to predict their quantum properties. The data partition we use has 110k, 10k, and 11k molecules in training, validation and testing sets. We minimize mean absolute error (MAE) between prediction and normalized ground truth.

**Training Details.** Please refer to Sec. D.1 in appendix for details on architecture, hyper-parameters and training time.

**Results.** We summarize the comparison to previous models in Table 1. Equiformer achieves overall better results across 12 regression tasks compared to each individual model. The comparison to SEGNN, which uses irreps features as Equiformer, demonstrates the effectiveness of combining non-lienar message passing with MLP attention. Additionally, Equiformer achieves better results for most tasks when compared to other equivariant Transformers, which are SE(3)-Transformer, TorchMD-NET and EQGAT. This demonstrates a better adaption of Transformers to 3D graphs and the effectiveness of the proposed equivariant graph attention. We note that for the tasks of $\mu$ and $R^2$, PaiNN and TorchMD-NET use different architectures, which take into account the property of the task. In contrast, we use the same architecture for all tasks. We compare training time in Sec. D.3.

### 5.2 MD17

**Dataset.** The MD17 dataset (Chmiela et al., 2017; Schütt et al., 2017; Chmiela et al., 2018) (CC BY-NC) consists of molecular dynamics simulations of small organic molecules, and the goal is to predict their energy and forces. We use 950 and 50 different configurations for training and validation sets and the rest for the testing set. Forces are derived as the negative gradient of energy with respect to atomic positions. We minimize MAE between prediction and normalized ground truth.

**Training Details.** Please refer to Sec. E.1 in appendix for details on architecture, hyper-parameters and training time.

**Results.** We train Equiformer with $L_{max} = 2$ and 3 and summarize the results in Table 2. Equiformer achieves overall better results across 8 molecules compared to each individual model. Compared to TorchMD-NET, which is also an equivariant Transformer, the difference lies in the proposed equivariant graph attention, which is more expressive and can support vectors of higher degree

| | Energy MAE (eV) ↓ | | | | | EwT (%) ↑ | | | | |
|---|---|---|---|---|---|---|---|---|---|---|
| Methods | ID | OOD Ads | OOD Cat | OOD Both | Average | ID | OOD Ads | OOD Cat | OOD Both | Average |
| CGCNN (Xie & Grossman, 2018) | 0.6149 | 0.9155 | 0.6219 | 0.8511 | 0.7509 | 3.40 | 1.93 | 3.10 | 2.00 | 2.61 |
| SchNet (Schütt et al., 2017) | 0.6387 | 0.7342 | 0.6616 | 0.7037 | 0.6846 | 2.96 | 2.33 | 2.94 | 2.21 | 2.61 |
| DimeNet++ (Gasteiger et al., 2020a) | 0.5621 | 0.7252 | 0.5756 | 0.6613 | 0.6311 | 4.25 | 2.07 | 4.10 | 2.41 | 3.21 |
| PaiNN (Schütt et al., 2021) | 0.575 | 0.783 | 0.604 | 0.743 | 0.6763 | 3.46 | 1.97 | 3.46 | 2.28 | 2.79 |
| SpinConv (Shuaibi et al., 2021) | 0.5583 | 0.7230 | 0.5687 | 0.6738 | 0.6310 | 4.08 | 2.26 | 3.82 | 2.33 | 3.12 |
| SphereNet (Liu et al., 2022) | 0.5625 | 0.7033 | 0.5708 | 0.6378 | 0.6186 | 4.47 | 2.29 | 4.09 | 2.41 | 3.32 |
| SEGNN (Brandstetter et al., 2022) | 0.5327 | 0.6921 | 0.5369 | 0.6790 | 0.6101 | **5.37** | **2.46** | **4.91** | 2.63 | **3.84** |
| Equiformer | **0.5037** | **0.6881** | **0.5213** | **0.6301** | **0.5858** | 5.14 | 2.41 | 4.67 | **2.69** | 3.73 |

Table 3: **Results on OC20 IS2RE testing set.**

| | Energy MAE (eV) ↓ | | | | | EwT (%) ↑ | | | | |
|---|---|---|---|---|---|---|---|---|---|---|
| Methods | ID | OOD Ads | OOD Cat | OOD Both | Average | ID | OOD Ads | OOD Cat | OOD Both | Average |
| GNS (Godwin et al., 2022) | 0.54 | 0.65 | 0.55 | 0.59 | 0.5825 | - | - | - | - | - |
| GNS + Noisy Nodes (Godwin et al., 2022) | 0.47 | 0.51 | 0.48 | 0.46 | 0.4800 | - | - | - | - | - |
| Graphormer (Shi et al., 2022) | 0.4329 | 0.5850 | 0.4441 | 0.5299 | 0.4980 | - | - | - | - | - |
| Equiformer | 0.4222 | 0.5420 | 0.4231 | 0.4754 | 0.4657 | 7.23 | 3.77 | 7.13 | 4.10 | 5.56 |
| Equiformer + Noisy Nodes | **0.4156** | **0.4976** | **0.4165** | **0.4344** | **0.4410** | **7.47** | **4.64** | **7.19** | **4.84** | **6.04** |

Table 4: **Results on OC20 IS2RE validation set when IS2RS is adopted during training.**

| | Energy MAE (eV) ↓ | | | | | EwT (%) ↑ | | | | | Training time |
|---|---|---|---|---|---|---|---|---|---|---|---|
| Methods | ID | OOD Ads | OOD Cat | OOD Both | Average | ID | OOD Ads | OOD Cat | OOD Both | Average | (GPU-days) |
| GNS + Noisy Nodes (Godwin et al., 2022) | 0.4219 | 0.5678 | 0.4366 | **0.4651** | 0.4728 | **9.12** | **4.25** | 8.01 | **4.64** | **6.5** | 56 (TPU) |
| Graphormer (Shi et al., 2022)[†] | **0.3976** | 0.5719 | **0.4166** | 0.5029 | 0.4722 | 8.97 | 3.45 | **8.18** | 3.79 | 6.1 | 372 (A100) |
| Equiformer + Noisy Nodes | 0.4171 | **0.5479** | 0.4248 | 0.4741 | **0.4660** | 7.71 | 3.70 | 7.15 | 4.07 | 5.66 | **24** (A6000) |

Table 5: **Results on OC20 IS2RE testing set when IS2RS is adopted during training.** † denotes using ensemble of models trained on both IS2RE training and validation sets. In contrast, we use the same single Equiformer model in Table 4, which is trained only on the training set. Note that Equiformer achieves better results with much less computation.

$L$ (i.e., we use $L_{max} = 2$ and 3) instead of restricting to type-0 and type-1 vectors (i.e., $L_{max} = 1$). For the last three molecules, although Equiformer with $L_{max} = 2$ achieves lower force MAE but higher energy MAE, we can adjust the weights of energy loss and force loss so that Equiformer achieves lower MAE for both energy and forces as shown in Table 12 in appendix. Compared to NequIP, which also uses irreps features and $L_{max} = 3$, Equiformer with $L_{max} = 2$ achieves overall lower MAE although including higher $L_{max}$ can improve performance. When using $L_{max} = 3$, Equiformer achieves lower MAE results for most molecules. This suggests that the proposed attention can improve upon linear messages even when the size of training sets becomes small. We compare the training time of NequIP and Equiformer in Sec. E.3. Additionally, for Equiformer, increasing $L_{max}$ from 2 to 3 improves MAE for most molecules except benzene, which results from overfitting.

### 5.3 OC20

**Dataset.** The Open Catalyst 2020 (OC20) dataset (Chanussot* et al., 2021) (Creative Commons Attribution 4.0 License) consists of larger atomic systems, each composed of a molecule called adsorbate placed on a slab called catalyst. Each input contains more atoms and more diverse atom types than QM9 and MD17. We focus on the task of initial structure to relaxed energy (IS2RE), which is to predict the energy of a relaxed structure (RS) given its initial structure (IS). Performance is measured in MAE and energy within threshold (EwT), the percentage in which predicted energy is within 0.02 eV of ground truth energy. In validation and testing sets, there are four sub-splits containing in-distribution adsorbates and catalysts (ID), out-of-distribution adsorbates (OOD-Ads), out-of-distribution catalysts (OOD-Cat), and out-of-distribution adsorbates and catalysts (OOD-Both). Please refer to Sec. F.1 for the detailed description of OC20 dataset.

**Setting.** We consider two training settings based on whether a node-level auxiliary task (Godwin et al., 2022) is adopted. In the first setting, we minimize MAE between predicted energy and ground truth energy without any node-level auxiliary task. In the second setting, we incorporate the task of initial structure to relaxed structure (IS2RS) as a node-level auxiliary task.

**Training Details.** Please refer to Sec. F.2 in appendix for details on Equiformer architecture, hyper-parameters and training time.

**IS2RE Results without Node-Level Auxiliary Task.** We summarize the results on validation and testing sets under the first setting in Table 15 in appendix and Table 3. Compared with state-of-the-art models like SEGNN and SphereNet, Equiformer consistently achieves the lowest MAE for all the four sub-splits in validation and testing sets. Note that EwT considers only the percentage of predictions close enough to ground truth and the distribution of errors, and therefore improvement in average errors (MAE) would not necessarily reflect that in error distributions (EwT). A more detailed

| | | Methods | | | | | | | | | | |
|---|---|---|---|---|---|---|---|---|---|---|---|---|
| Index | Non-linear message passing | MLP attention | Dot product attention | Task Unit | $\alpha$ $a_0^3$ | $\Delta\varepsilon$ meV | $\varepsilon_{\text{HOMO}}$ meV | $\varepsilon_{\text{LUMO}}$ meV | $\mu$ D | $C_\nu$ cal/mol K | Training time (minutes/epoch) | Number of parameters |
| 1 | ✓ | ✓ | | | .046 | 30 | 15 | 14 | .011 | .023 | 12.1 | 3.53M |
| 2 | | ✓ | | | .051 | 32 | 16 | 16 | .013 | .025 | 7.2 | 3.01M |
| 3 | | | ✓ | | .053 | 32 | 17 | 16 | .013 | .025 | 7.8 | 3.35M |

Table 6: **Ablation study results on QM9.**

| | | Methods | | Energy MAE (eV) ↓ | | | | | EwT (%) ↑ | | | | | | |
|---|---|---|---|---|---|---|---|---|---|---|---|---|---|---|---|
| Index | Non-linear message passing | MLP attention | Dot product attention | ID | OOD Ads | OOD Cat | OOD Both | Average | ID | OOD Ads | OOD Cat | OOD Both | Average | Training time (minutes/epoch) | Number of parameters |
| 1 | ✓ | ✓ | | 0.5088 | 0.6271 | 0.5051 | 0.5545 | 0.5489 | 4.88 | 2.93 | 4.92 | 2.98 | 3.93 | 130.8 | 9.12M |
| 2 | | ✓ | | 0.5168 | 0.6308 | 0.5088 | 0.5657 | 0.5555 | 4.59 | 2.82 | 4.79 | 3.02 | 3.81 | 91.2 | 7.84M |
| 3 | | | ✓ | 0.5386 | 0.6382 | 0.5297 | 0.5692 | 0.5689 | 4.37 | 2.60 | 4.36 | 2.86 | 3.55 | 99.3 | 8.72M |

Table 7: **Ablation study results on OC20 IS2RE validation set.**

discussion can be found in Sec. F.6 in appendix. We also note that models are trained by minimizing MAE, and therefore comparing MAE could mitigate the discrepancy between training objectives and evaluation metrics and that OC20 leaderboard ranks the relative performance according to MAE. Additionally, we compare the training time of SEGNN and Equiformer in Sec. F.5.

**IS2RE Results with IS2RS Node-Level Auxiliary Task.** We report the results on validation and testing sets in Table 4 and Table 5. As of the date of the submission of this work, Equiformer achieves the best results on IS2RE task when only IS2RE and IS2RS data are used. Notably, the result in Table 5 is achieved with much less computation. We note that under this setting, greater depths and thus more computation translate to better performance (Godwin et al., 2022) and that Equiformer demonstrates incorporating equivariant features and the proposed equivariant graph attention can improve training efficiency by $2.3\times$ to $15.5\times$ compared to invariant message passing networks and invariant Transformers.

### 5.4 Ablation Study

We conduct ablation studies to show that Equiformer with dot product attention and linear message passing has already achieved strong empirical results and demonstrate the improvement brought by MLP attention and non-linear messages in the proposed equivariant graph attention. Dot product (DP) attention only differs from MLP attention in how attention weights $a_{ij}$ are generated from $f_{ij}$. Please refer to Sec. C.3 in appendix for further details. For experiments on QM9 and OC20, unless otherwise stated, we follow the hyper-parameters used in previous experiments.

**Result on QM9.** The comparison is summarized in Table 6. Compared with models in Table 1, Equiformer with dot product attention and linear message passing (Index 3) achieves competitve results. Non-linear messages improve upon linear messages when MLP attention is used while non-linear messages increase the number of tensor product operations in each block from 1 to 2 and thus inevitably increase training time. On the other hand, MLP attention achieves similar results to DP attention. We conjecture that DP attention with linear operations is expressive enough to capture common attention patterns as the numbers of nighboring nodes and atom species are much smaller than those in OC20. However, MLP attention is roughly $8\%$ faster as it directly generates scalar features and attention weights from $f_{ij}$ instead of producing additional key and query irreps features for attention weights.

**Result on OC20.** We consider the setting of training without auxiliary task and summarize the comparison in Table 7. Compared with models in Table 15, Equiformer with dot product attention and linear message passing (Index 3) has already outperformed all previous models. Non-linear messages consistently improve upon linear messages. In contrast to the results on QM9, MLP attention achieves better performance than DP attention and is $8\%$ faster. We surmise this is because OC20 contains larger atomistic graphs with more diverse atom species and therefore requires more expressive attention mechanisms. Note that Equiformer can potentially improve upon previous equivariant Transformers (Fuchs et al., 2020; Thölke & Fabritiis, 2022; Le et al., 2022) since they use less expressive attention mechanisms similar to Index 3 in Table 7.

### 6 Conclusion

In this work, we propose Equiformer, a graph neural network (GNN) combining the strengths of Transformers and equivariant features based on irreducible representations (irreps). With irreps features, we build upon existing generic GNNs and Transformer networks by incorporating equivariant operations like tensor products. We further propose equivariant graph attention, which incorporates multi-layer perceptron attention and non-linear messages. Experiments on QM9, MD17 and OC20 demonstrate the effectiveness of Equiformer and ablation studies show the improvement of the proposed equivariant graph attention over typical attention in Transformers.

## 7 ETHICS STATEMENT

Equiformer achieves more accurate approximations of quantum properties calculation. We believe there is much more to be gained by harnessing these abilities for productive investigation of molecules and materials relevant to application such as energy, electronics, and pharmaceuticals, than to be lost by applying these methods for adversarial purposes like creating hazardous chemicals. Additionally, there are still substantial hurdles to go from the identification of a useful or harmful molecule to its large-scale deployment.

Moreover, we discuss several limitations of Equiformer and the proposed equivariant graph attention in Sec. G in appendix.

## 8 REPRODUCIBILITY STATEMENT

We include details on architectures, hyper-parameters and training time in Sec. D.1 (QM9), Sec. E.1 (MD17) and Sec. F.2 (OC20).

The code for reproducing the results of Equiformer on QM9, MD17 and OC20 datasets is available at https://github.com/atomicarchitects/equiformer.

## ACKNOWLEDGEMENT

We thank Simon Batzner, Albert Musaelian, Mario Geiger, Johannes Brandstetter, and Rob Hesselink for helpful discussions including help with the OC20 dataset. We also thank the e3nn (Geiger et al., 2022) developers and community for the library and detailed documentation. We acknowledge the MIT SuperCloud and Lincoln Laboratory Supercomputing Center (Reuther et al., 2018) for providing high performance computing and consultation resources that have contributed to the research results reported within this paper.

Yi-Lun Liao and Tess Smidt were supported by DOE ICDI grant DE-SC0022215.

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

APPENDIX

## A ADDITIONAL BACKGROUND

In this section, we provide additional mathematical background helpful for the discussion of the proposed method. Other works (Thomas et al., 2018; Weiler et al., 2018; Kondor et al., 2018; Anderson et al., 2019; Fuchs et al., 2020; Brandstetter et al., 2022) also provide similar background. We encourage interested readers to see these works (Zee, 2016; Dresselhaus et al., 2007) for more in-depth and pedagogical presentations.

### A.1 GROUP THEORY

**Definition of Groups.** A group is an algebraic structure that consists of a set $G$ and a binary operator $\circ : G \times G \to G$ and is typically denoted as $G$. Groups satisfy the following four axioms:

1. Closure: $g \circ h \in G$ for all $g, h \in G$.
2. Identity: There exists an identity element $e \in G$ such that $g \circ e = e \circ g = g$ for all $g \in G$.

3. Inverse: For each $g \in G$, there exists an inverse element $g^{-1} \in G$ such that $g \circ g^{-1} = g^{-1} \circ g = e$.

4. Associativity: $f \circ g \circ h = (f \circ g) \circ h = f \circ (g \circ h)$ for all $f, g, h \in G$.

In this work, we focus on 3D rotation, translation and inversion. Relevant groups include:

1. The Euclidean group in three dimensions $E(3)$: 3D rotation, translation and inversion.

2. The special Euclidean group in three dimensions $SE(3)$: 3D rotation and translation.

3. The orthogonal group in three dimensions $O(3)$: 3D rotation and inversion.

4. The special orthogonal group in three dimensions $SO(3)$: 3D rotation.

**Group Representations.** The actions of groups define transformations. Formally, a transformation acting on vector space $X$ parametrized by group element $g \in G$ is an injective function $T_g : X \to X$. A powerful result of group representation theory is that these transformations can be expressed as matrices which act on vector spaces via matrix multiplication. These matrices are called the group representations. Formally, a group representation $D : G \to GL(N)$ is a mapping between a group $G$ and a set of $N \times N$ invertible matrices. The group representation $D(g) : X \to X$ maps an $N$-dimensional vector space $X$ onto itself and satisfies $D(g)D(h) = D(g \circ h)$ for all $g, h \in G$.

How a group is represented depends on the vector space it acts on. If there exists a change of basis $P$ in the form of an $N \times N$ matrix such that $P^{-1}D(g)P = D'(g)$ for all $g \in G$, then we say the two group representations are equivalent. If $D'(g)$ is block diagonal, which means that $g$ acts on independent subspaces of the vector space, the representation $D(g)$ is reducible. A particular class of representations that are convenient for composable functions are irreducible representations or "irreps", which cannot be further reduced. We can express any group representation of $SO(3)$ as a direct sum (concatenation) of irreps (Zee, 2016; Dresselhaus et al., 2007; Geiger et al., 2022):

$$D(g) = P^{-1}\left(\bigoplus_i D_{l_i}(g)\right)P = P^{-1}\begin{pmatrix} D_{l_0}(g) & & \\ & D_{l_1}(g) & \\ & & \ddots \end{pmatrix}P \qquad (6)$$

where $D_{l_i}(g)$ are Wigner-D matrices with degree $l_i$ as metnioned in Sec. 3.2.

## A.2 EQUIVARIANCE

**Definition of Equivariance and Invariance.** Equivariance is a property of a function $f : X \to Y$ mapping between vector spaces $X$ and $Y$. Given a group $G$ and group representations $D_X(g)$ and $D_Y(g)$ in input and output spaces $X$ and $Y$, $f$ is equivariant to G if $D_Y(g)f(x) = f(D_X(g)x)$ for all $x \in X$ and $g \in G$. Invariance corresponds to the case where $D_Y(g)$ is the identity $I$ for all $g \in G$.

**Equivariance in Neural Networks.** Group equivariant neural networks are guaranteed to to make equivariant predictions on data transformed by a group. Additionally, they are found to be data-efficient and generalize better than non-symmetry-aware and invariant methods (Batzner et al., 2022; Rackers et al., 2023; Frey et al., 2022). For 3D atomistic graphs, we consider equivariance to the Euclidean group $E(3)$, which consists of 3D rotation, translation and inversion. For translation, we operate on relative positions and therefore our networks are invariant to 3D translation. We achieve equivariance to rotation and inversion by representing our input data, intermediate features and outputs in vector spaces of $O(3)$ irreps and acting on them with only equivariant operations.

## A.3 EQUIVARIANT FEATURES BASED ON VECTOR SPACES OF IRREDUCIBLE REPRESENTATIONS

**Irreducible Representations of Inversion.** The group of inversion $\mathbb{Z}_2$ only has two elements, identity and inversion, and two irreps, even $e$ and odd $o$. Vectors transformed by irrep $e$ do not change sign under inversion while those by irrep $o$ do. We create irreps of $O(3)$ by simply multiplying those of $SO(3)$ and $\mathbb{Z}_2$ and introduce parity $p$ to type-$L$ vectors to denote how they transform under inversion. Thus, type-$L$ vectors in $SO(3)$ become type-$(L, p)$ vectors in $O(3)$, where $p$ is $e$ or $o$.

**Irreps Features.** As discussed in Sec. 3.2 in the main text, we use type-$L$ vectors for $SE(3)$-equivariant irreps features[1] and type-$(L, p)$ vectors for $E(3)$-equivariant irreps features. Parity $p$ denotes whether vectors change sign under inversion and can be either $e$ (even) or $o$ (odd). Vectors with $p = o$ change sign under inversion while those with $p = e$ do not. Scalar features correspond to type-$0$ vectors in the case of $SE(3)$-equivariance and correspond to type-$(0, e)$ in the case of $E(3)$-equivariance whereas type-$(0, o)$ vectors correspond to pseudo-scalars. Euclidean vectors in $\mathbb{R}^3$ correspond to type-$1$ vectors and type-$(1, o)$ vectors whereas type-$(1, e)$ vectors correspond to pseudo-vectors. Note that type-$(L, e)$ vectors and type-$(L, o)$ vectors are considered vectors of different types in equivariant linear layers and layer normalizations.

**Spherical Harmonics.** Euclidean vectors $\vec{r}$ in $\mathbb{R}^3$ can be projected into type-$L$ vectors $f^{(L)}$ by using spherical harmonics $Y^{(L)}$: $f^{(L)} = Y^{(L)}(\frac{\vec{r}}{||\vec{r}||})$ (Smidt et al., 2021). This is equivalent to the Fourier transform of the angular degree of freedom $\frac{\vec{r}}{||\vec{r}||}$, which can be optionally weighted by $||\vec{r}||$. In the case of $SE(3)$-equivariance, $f^{(L)}$ transforms in the same manner as type-$L$ vectors. For $E(3)$-equivariance, $f^{(L)}$ behaves as type-$(L, p)$ vectors, where $p = e$ if $L$ is even and $p = o$ if $L$ is odd. Visualization of spherical harmonics can be found in this website.

**Vectors of Higher $L$ and Other Parities.** Although previously we have restricted concrete examples of vector spaces of $O(3)$ irreps to commonly encountered scalars (type-$(0, e)$ vectors) and Euclidean vectors (type-$(1, o)$ vectors), vector of higher $L$ and other parities are equally physical. For example, the moment of inertia (how an object rotates under torque) transforms as a $3 \times 3$ symmetric matrix, which has symmetric-traceless components behaving as type-$(2, e)$ vectors. Elasticity (how an object deforms under loading) transforms as a rank-4 or $3 \times 3 \times 3 \times 3$ symmetric tensor, which includes components acting as type-$(4, e)$ vectors.

## A.4 TENSOR PRODUCT

**Tensor Product for $O(3)$.** We use tensor products to interact different type-$(L, p)$ vectors. We extend our discussion in Sec. 3.3 in the main text to include inversion and type-$(L, p)$ vectors. The tensor product denoted as $\otimes$ uses Clebsch-Gordan coefficients to combine type-$(L_1, p_1)$ vector $f^{(L_1, p_1)}$ and type-$(L_2, p_2)$ vector $g^{(L_2, p_2)}$ and produces type-$(L_3, p_3)$ vector $h^{(L_3, p_3)}$ as follows:

$$h_{m_3}^{(L_3, p_3)} = (f^{(L_1, p_1)} \otimes g^{(L_2, p_2)})_{m_3} = \sum_{m_1=-L_1}^{L_1} \sum_{m_2=-L_2}^{L_2} C_{(L_1, m_1)(L_2, m_2)}^{(L_3, m_3)} f_{m_1}^{(L_1, p_1)} g_{m_2}^{(L_2, p_2)} \quad (7)$$

$$p_3 = p_1 \times p_2 \quad (8)$$

The only difference of tensor products for $O(3)$ as described in Eq. 7 from those for $SO(3)$ described in Eq. 1 is that we additionally keep track of the output parity $p_3$ as in Eq. 8 and use the following multiplication rules: $e \times e = e$, $o \times o = e$, and $e \times o = o \times e = o$. For example, the tensor product of a type-$(1, o)$ vector and a type-$(1, e)$ vector can result in one type-$(0, o)$ vector, one type-$(1, o)$ vector, and one type-$(2, o)$ vector.

**Clebsch-Gordan Coefficients.** The Clebsch-Gordan coefficients for $SO(3)$ are computed from integrals over the basis functions of a given irreducible representation, e.g., the real spherical harmonics, as shown below and are tabulated to avoid unnecessary computation.

$$C_{(L_1, m_1)(L_2, m_2)}^{(L_3, m_3)} = |L_1 m_1; L_2 m_2\rangle \langle L_3 m_3| = \int d\Omega\, Y_{m_1}^{(L_1)*}(\Omega) Y_{m_2}^{(L_2)*}(\Omega) Y_{m_3}^{(L_3)}(\Omega) \quad (9)$$

For many combinations of $L_1$, $L_2$, and $L_3$, the Clebsch-Gordan coefficients are zero. The gives rise to the following selection rule for non-trivial coefficients: $-|L_1 + L_2| \leq L_3 \leq |L_1 + L_2|$.

---

[1]In SEGNN (Brandstetter et al., 2022), they are also referred to as steerable features. We use the term "irreps features" to remain consistent with e3nn (Geiger et al., 2022) library.

**Examples of Tensor Products.** Tensor products generally define the interaction between different type-$(L, p)$ vectors in a symmetry-preserving manner and consist of common operations as follows:

1. Scalar-scalar multiplication: scalar ($L = 0, p = e$) $\otimes$ scalar ($L = 0, p = e$) $\rightarrow$ scalar ($L = 0, p = e$).

2. Scalar-vector multiplication: scalar ($L = 0, p = e$) $\otimes$ vector ($L = 1, p = o$) $\rightarrow$ vector ($L = 1, p = o$).

3. Vector dot product: vector ($L = 1, p = o$) $\otimes$ vector ($L = 1, p = o$) $\rightarrow$ scalar ($L = 0, p = e$).

4. Vector cross product: vector ($L = 1, p = o$) $\otimes$ vector ($L = 1, p = o$) $\rightarrow$ pseudo-vector ($L = 1, p = e$).

## B  RELATED WORKS

### B.1  GRAPH NEURAL NETWORKS FOR 3D ATOMISTIC GRAPHS

Graph neural networks (GNNs) are well adapted to perform property prediction of atomic systems because they can handle discrete and topological structures. There are two main ways to represent atomistic graphs (Townshend et al., 2021), which are chemical bond graphs, sometimes denoted as 2D graphs, and 3D spatial graphs. Chemical bond graphs use edges to represent covalent bonds without considering 3D geometry. Due to their similarity to graph structures in other applications, generic GNNs (Hamilton et al., 2017; Gilmer et al., 2017; Kipf & Welling, 2017; Xu et al., 2019; Veličković et al., 2018; Brody et al., 2022) can be directly applied to predict their properties (Ruddigkeit et al., 2012; Ramakrishnan et al., 2014; Ramsundar et al., 2019; Hu et al., 2020; 2021). On the other hand, 3D spatial graphs consider positions of atoms in 3D spaces and therefore 3D geometry. Although 3D graphs can faithfully represent atomistic systems, one challenge of moving from chemical bond graphs to 3D spatial graphs is to remain invariant or equivariant to geometric transformation acting on atom positions. Therefore, invariant neural networks and equivariant neural networks have been proposed for 3D atomistic graphs, with the former leveraging invariant information like distances and angles and the latter operating on geometric tensors like type-$L$ vectors.

### B.2  DETAILED COMPARISON BETWEEN EQUIVARIANT TRANSFORMERS

First, we compare the impact of previous equivariant Transformers (Fuchs et al., 2020; Thölke & Fabritiis, 2022; Le et al., 2022) in the following four aspects:

1. Previous equivariant Transformers do not perform well across datasets. For instance, SE(3)-Transformer (Fuchs et al., 2020) is not as performant as other equivariant networks on QM9 as shown in Table 1. TorchMD-NET (Thölke & Fabritiis, 2022) does not achieve comparable results to NequIP (Batzner et al., 2022) on MD17 as shown in Table 2 although it is competitive on QM9 in Table 1.

2. Equiformer simultaneously achieves the best results for MD17, QM9 and OC20 datasets, indicating that the Transformer architecture is generally effective in the literature of equivariant neural networks and 3D atomistic graphs.

3. Extensive ablation studies have been conducted to justify a better attention mechanism in this literature.

4. To the best of our knowledge, we are the first to apply equivariant Transformers to large and complicated datasets like OC20 and demonstrate that equivariant Transformers can achieve competitive results to large models like GNS (Godwin et al., 2022) and Graphormer (Shi et al., 2022) while saving $2.3\times$ to $15.5\times$ training time as summarized in Table 5.

Second, we compare the technical differences of architectures of equivariant Transformers. The proposed Equiformer consists of equivariant graph attention and an equivariant Transformer architecture. The latter is obtained by simply replacing orginal operations in Transformers with their equivariant counterparts and including tensor product operations and corresponds to "Equiformer with dot product attention and linear message passing" as indicated by Index 3 in Table 6 and 7.

Although with minimal modifications to original Transformers, we note that this architecture has not been explored in previous equivariant Transformers and achieves competitive results on QM9 and OC20 datasets. Below we discuss the advantages of the architecture, Equiformer with dot product attention and linear message passing, over other equivariant Transformers:

1. **Simpler.** We simply replace original operations with their equivariant counterparts and include tensor products without making further modifications. In contrast, SE(3)-Transformer (Fuchs et al., 2020) merges normalization and activation to form norm nonlinearities, which does not exist in original Transformers. We empirically find that the norm nonlinearities (Fuchs et al., 2020) leads to higher errors compared to the equivariant layer norm used by our work.

2. **More general.** SE(3)-Transformer (Fuchs et al., 2020) and our proposed architecture can support vectors of any degree $L$ while other equivariant Transformers (Thölke & Fabritiis, 2022; Le et al., 2022) are limited to $L = 0$ and 1. It has been shown that higher $L$ (e.g., $L$ up to 2 and 3) can improve the performance of networks on QM9 (Brandstetter et al., 2022) and MD17 (Batzner et al., 2022). Thus, the incapability to use $L$ higher than 1 can limit their performance.

3. **More efficient tensor products.** Compared to SE(3)-Transformer (Fuchs et al., 2020), we use more efficient depth-wise tensor products instead of fully connected tensor products. Since the proposed architecture and SE(3)-Transformer (Fuchs et al., 2020) use relative distances to parametrize the weights of tensor products, depth-wise tensor products enalbe using more channels without incurring out-of-memory errors. Specifically, for QM9, SE(3)-Transformer (Fuchs et al., 2020) only uses 16 channels for vectors of each degree $L$ while the proposed architecture can use 128, 64, and 32 channels for vectors of degree 0, 1, and 2. Using a very small number of channels can potentially lead to insufficient model capacity.

We note that the novelty of the proposed architecture, Equiformer with dot product attention and linear message passing, lies in how we choose the right operations as well as internal representations (i.e., vectors of any degree $L$) and combine them in an effective manner that achieves the three advantages mentioned above. We further improve this simple architecture with our proposed equivariant graph attention, which consists of MLP attention and non-linear message passing.

## B.3 INVARIANT GNNS

Previous works (Schütt et al., 2017; Xie & Grossman, 2018; Unke & Meuwly, 2019; Gasteiger et al., 2020b;a; Qiao et al., 2020; Liu et al., 2022; Shuaibi et al., 2021; Klicpera et al., 2021) extract invariant information from 3D atomistic graphs and operate on the resulting invariant graphs. They mainly differ in leveraging different geometric information such as distances, bond angles (3 atom features) or dihedral angles (4 atom features). SchNet (Schütt et al., 2017) uses relative distances and proposes continuous-filter convolutional layers to learn local interaction between atom pairs. DimeNet series (Gasteiger et al., 2020b;a) incorporate bond angles by using triplet representations of atoms. SphereNet (Liu et al., 2022) and GemNet (Klicpera et al., 2021; Gasteiger et al., 2022) further extend to consider dihedral angles for better performance. In order to consider directional information contained in angles, they rely on triplet or quadruplet representations of atoms. In addition to being memory-intensive (Sriram et al., 2022), they also change graph structures by introducing higher-order interaction terms (Chen et al., 2019), which would require non-trivial modifications to generic GNNs in order to apply them to 3D graphs. In contrast, the proposed Equiformer uses equivariant irreps features to consider directional information without complicating graph structures and therefore can directly inherit the design of generic GNNs.

## B.4 ATTENTION AND TRANSFORMER

**Graph Attention.** Graph attention networks (GAT) (Veličković et al., 2018; Brody et al., 2022) use multi-layer perceptrons (MLP) to calculate attention weights in a similar manner to message passing networks. Subsequent works using graph attention mechanisms follow either GAT-like MLP attention (Busbridge et al., 2019; Kim & Oh, 2021) or Transformer-like dot product attention (Zhang et al., 2018a; Gao & Ji, 2019; Shi et al., 2020; Dwivedi & Bresson, 2020; Kim & Oh, 2021; Kreuzer et al., 2021). In particular, Kim *et al.* (Kim & Oh, 2021) compares these two types of attention

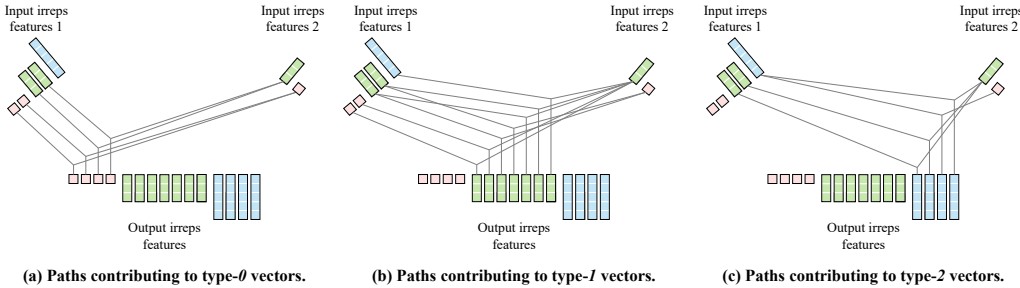

(a) Paths contributing to type-*0* vectors.  (b) Paths contributing to type-*1* vectors.  (c) Paths contributing to type-*2* vectors.

Figure 3: **An alternative visualization of the depth-wise tensor product.** We follow the visualization of tensor products in `e3nn` (Geiger et al., 2022) and separate paths into three parts based on the types of output vectors. We note that one vector in the output irreps feature depends only on one vector in each input irreps feature.

mechanisms empirically under a self-supervised setting. Brody *et al.* (Brody et al., 2022) analyzes their theoretical differences and compares their performance in general settings.

**Graph Transformer.** A different line of research focuses on adapting standard Transformer networks to graph problems (Dwivedi & Bresson, 2020; Rong et al., 2020; Kreuzer et al., 2021; Ying et al., 2021; Shi et al., 2022). They adopt dot product attention in Transformers (Vaswani et al., 2017) and propose different approaches to incorporate graph-related inductive biases into their networks. GROVE (Rong et al., 2020) includes additional message passing layers or graph convolutional layers to incorporate local graph structures when calculating attention weights. SAN (Kreuzer et al., 2021) proposes to learn position embeddings of nodes with full Laplacian spectrum. Graphormer (Ying et al., 2021) proposes to encode degree information in centrality embeddings and encode distances and edge features in attention biases. The proposed Equiformer belongs to one of these attempts to generalize standard Transformers to graphs and is dedicated to 3D graphs. To incorporate 3D-related inductive biases, we adopt an equivariant version of Transformers with irreps features and propose novel equivariant graph attention.

## C    DETAILS OF ARCHITECTURE

### C.1    EQUIVARIANT OPERATION USED IN EQUIFORMER

We illustrate the equivariant operations used in Equiformer in Fig. 2 and provide an alternative visualization of depth-wise tensor products in Fig. 3.

Besides, we analyze how each equivariant operation remains equivariant and satisfies that $f(D_X(g)x) = D_Y(g)f(x)$, where $f$ is a function mapping between vector spaces $X$ and $Y$, and $D_X(g)$ and $D_Y(g)$ are transformation matrices parametrized by $g$ in $X$ and $Y$.

1. **Linear.** Since for each degree $L$, one output type-$L$ vector is a linear combination of other input type-$L$ vectors, which are transformed by the same matrix $D_X(g)$, the output type-$L$ vector is transformed by the same matrix, meaning that $D_X(g) = D_Y(g)$.

2. **Layer normalization.** For scalar parts ($L = 0$), they are always the same regardless of $E(3)$ transformations, and thus we can apply any function to them. For non-scalar parts ($L > 0$), the L2-norm of any type-$L$ vector is invariant to $E(3)$ transformations. Therefore, the scaling of dividing by the root mean square value (RMS) of L2-norm and multiplying by a learnable parameter $\gamma$ remains the same under $E(3)$ transformations. Multiplying an equivariant feature with an invariant number results in an equivariant feature, and therefore the operation is equivariant.

3. **Gate.** Similar to layer normalization, we can apply any function to the scalar part ($L = 0$). We apply SiLU to the first $C_0$ channels of the scalar part and sigmoid to other channels to obtain non-linear weights. For the non-scalar part ($L > 0$), we multiply each type-$L$

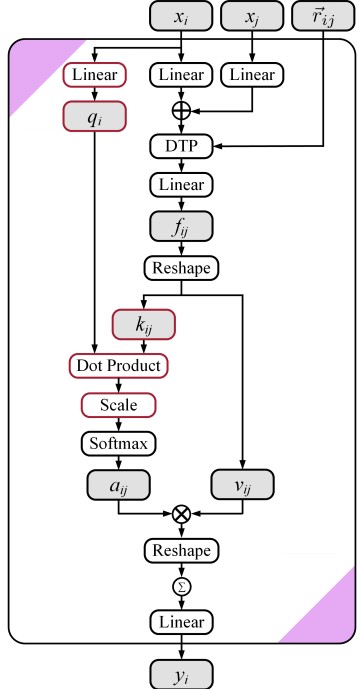

Figure 4: **Architecture of equivariant dot product attention without non-linear message passing.** In this figure, "⊗" denotes multiplication, "⊕" denotes addition, and "DTP" stands for depth-wise tensor product. $\sum$ within a circle denotes summation over all neighbors. Gray cells indicate intermediate irreps features. We highlight the difference of dot product attention from multi-layer perceptron attention in red. Note that key $k_{ij}$ and value $v_{ij}$ are irreps features and therefore $f_{ij}$ in dot product attention typically has more channels than that in multi-layer perceptron attention.

vector with its corresponding non-linear weight. Since the non-linear weights are invariant, multiplying equivariant features with those non-linear weights results in equivariant features.

4. **Depth-wise tensor product.** This operation is based on equivariant tensor product operations and restricts that one channel in output irreps feature depends on one channel in input irreps features. The one-to-one dependence of channels does not change the interaction of different type-$L$ vectors, and therefore, the operation is equivariant.

## C.2 EQUIFORMER ARCHITECTURE

For simplicity and because most works we compare with do not include equivariance to inversion, we adopt $SE(3)$-equivariant irreps features in Equiformer for experiments in the main text and note that $E(3)$-equivariant irreps features can be easily incorporated into Equiformer.

We define architectural hyper-parameters like the number of channels in some layers in Equiformer, which are used to specify the detailed architectures in Sec. D.1, Sec. E.1 and Sec. F.2.

We use $d_{embed}$ to denote embedding dimension, which defines the dimension of most irreps features. Specifically, all irreps features $x_i, y_i$ in Fig. 1 have dimension $d_{embed}$ unless otherwise stated. Besides, we use $d_{sh}$ to represent the dimension of spherical harmonics embeddings of relative positions in all depth-wise tensor products.

For equivariant graph attention in Fig. 1(b), the first two linear layers have the same output dimension $d_{embed}$. The output dimension of depth-wise tensor products (DTP) are determined by that of input irreps features. Equivariant graph attention consists of $h$ parallel attention functions, and the value vector in each attention function has dimension $d_{head}$. We refer to $h$ and $d_{head}$ as the number of heads and head dimension, respectively. By default, we set the number of channels in scalar feature $f_{ij}^{(0)}$ to be the same as the number of channels of type-0 or type-$(0, e)$ vectors in $v_{ij}$. When non-linear

messages are adopted in $v_{ij}$, we set the dimension of output irreps features in gate activation to be $h \times d_{head}$. Therefore, we can use two hyper-parameters $h$ and $d_{head}$ to specify the detailed architecture of equivariant graph attention.

As for feed forward networks (FFNs), we denote the dimension of output irreps features in gate activation as $d_{ffn}$. The FFN in the last Transformer block has output dimension $d_{feature}$, and we set $d_{ffn}$ of the last FFN, which is followed by output head, to be $d_{feature}$ as well. Thus, two hyper-parameters $d_{ffn}$ and $d_{feature}$ are used to specify architectures of FFNs and the output dimension after Transformer blocks.

Irreps features contain channels of vectors with degrees up to $L_{max}$. We denote $C_L$ type-$L$ vectors as $(C_L, L)$ and $C_{(L,p)}$ type-$(L, p)$ vectors as $(C_{(L,p)}, L, p)$ and use brackets to represent concatenations of vectors. For example, the dimension of irreps features containing 256 type-0 vectors and 128 type-1 vectors can be represented as $[(256, 0), (128, 1)]$.

## C.3 Dot Product Attention

We illustrate the dot product attention without non-linear message passing used in ablation study in Fig. 4. The architecture is adapted from SE(3)-Transformer (Fuchs et al., 2020). The difference from multi-layer perceptron attention lies in how we obtain attention weights $a_{ij}$ from $f_{ij}$. We split $f_{ij}$ into two irreps features, key $k_{ij}$ and value $v_{ij}$, and obtain query $q_i$ with a linear layer. Then, we perform scaled dot product (Vaswani et al., 2017) between $q_i$ and $k_{ij}$ for attention weights.

## C.4 Incorporating $E(3)$-Equivariance

To incorporate $E(3)$-equivariance to Equiformer, we can directly use the same architecture described in Fig. 1 with the following two modifications:

1. We change internal representations from type-$L$ vectors to type-$(L, p)$ vectors as mentioned in Sec. A.3. Note that type-$(L, e)$ vectors and type-$(L, o)$ vectors are considered different types by equivariant operations and that scalars correspond to only type-$(0, e)$ vectors and do not include type-$(0, o)$ vectors.

2. The behaviors of equivariant operations are changed accordingly since parities $p$ are included in internal representations. The operations of linear and layer normalization will treat type-$(L, e)$ vectors and type-$(L, o)$ vectors as different types. This means that we linearly combine or normalize these two types of vectors in a separate manner. In Sec. A.4, we discuss how tensor products behave when $E(3)$ equivariance is considered. For the operation of gate, we apply activation functions to type-$(0, e)$ vectors (scalars) and treat type-$(0, o)$ vectors (pseudo-scalars) in the same manner as vectors of higher $L$. Specifically, given input $x$ containing non-scalar $C_{(L,p)}$ type-$(L, p)$ vectors with $0 < L \le L_{max}$ and $p \in \{e, o\}$, $C_{(0,o)}$ type-$(0, o)$ vectors and $(C_{(0,e)} + C_{(0,o)} + \sum_{L=1}^{L_{max}} \sum_{p \in \{e,o\}} C_{(L,p)})$ type-$(0, e)$ vectors, we apply SiLU to the first $C_{(0,e)}$ type-$(0, e)$ vectors and sigmoid function to the other $(C_{(0,o)} + \sum_{L=1}^{L_{max}} \sum_{p \in \{e,o\}} C_{(L,p)})$ type-$(0, e)$ vectors to obtain non-linear weights and multiply each pseudo-scalar or type-$(L, p)$ vector with corresponding non-linear weights. After the gate activation, the number of channels for type-$(0, e)$ vectors is reduced to $C_{(0,e)}$.

## C.5 Discussion on Computational Complexity

We discuss the computational complexity of the proposed equivariant graph attention here.

First, we compare dot product attention with MLP attention when linear messages are used for value $v_{ij}$. Dot product attention requires taking the dot product of two irreps features, query $q_i$ and key $k_{ij}$, for attention weights, and both $q_i$ and $k_{ij}$ have the same dimension as value $v_{ij}$. In contrast, MLP attention uses only scalar features $f_{ij}^{(0)}$ for attention weights. The dimension of scalar features $f_{ij}^{(0)}$ is the same as that of the scalar part of $v_{ij}$. Therefore, MLP attention generates less and smaller intermediate features for attention weights and is faster than dot product attention.

| Hyper-parameters | Value or description |
|---|---|
| Optimizer | AdamW |
| Learning rate scheduling | Cosine learning rate with linear warmup |
| Warmup epochs | 5 |
| Maximum learning rate | $1.5 \times 10^{-4}$, $5 \times 10^{-4}$ |
| Batch size | 64, 128 |
| Number of epochs | 300, 600 |
| Weight decay | $0$, $5 \times 10^{-3}$ |
| Dropout rate | 0.0, 0.1, 0.2 |
| Cutoff radius (Å) | 5 |
| Number of radial bases | 128 for Gaussian radial basis, 8 for radial bessel basis |
| Hidden sizes of radial functions | 64 |
| Number of hidden layers in radial functions | 2 |
| Equiformer | |
| Number of Transformer blocks | 6 |
| Embedding dimension $d_{embed}$ | $[(128, 0), (64, 1), (32, 2)]$ |
| Spherical harmonics embedding dimension $d_{sh}$ | $[(1, 0), (1, 1), (1, 2)]$ |
| Number of attention heads $h$ | 4 |
| Attention head dimension $d_{head}$ | $[(32, 0), (16, 1), (8, 2)]$ |
| Hidden dimension in feed forward networks $d_{ffn}$ | $[(384, 0), (192, 1), (96, 2)]$ |
| Output feature dimension $d_{feature}$ | $[(512, 0)]$ |
| $E(3)$-Equiformer | |
| Number of Transformer blocks | 6 |
| Embedding dimension $d_{embed}$ | $[(128, 0, e), (32, 0, o), (32, 1, e), (32, 1, o), (16, 2, e), (16, 2, o)]$ |
| Spherical harmonics embedding dimension $d_{sh}$ | $[(1, 0, e), (1, 1, o), (1, 2, e)]$ |
| Number of attention heads $h$ | 4 |
| Attention head dimension $d_{head}$ | $[(32, 0, e), (8, 0, o), (8, 1, e), (8, 1, o), (4, 2, e), (4, 2, o)]$ |
| Hidden dimension in feed forward networks $d_{ffn}$ | $[(384, 0, e), (96, 0, o), (96, 1, e), (96, 1, o), (48, 2, e), (48, 2, o)]$ |
| Output feature dimension $d_{feature}$ | $[(512, 0, e)]$ |

Table 8: **Hyper-parameters for QM9 dataset.** We denote $C_L$ type-$L$ vectors as $(C_L, L)$ and $C_{(L,p)}$ type-$(L, p)$ vectors as $(C_{(L,p)}, L, p)$ and use brackets to represent concatenations of vectors.

Second, compared to linear messages, using non-linear messages increases the number of tensor product operations from 1 to 2. Since tensor products are compute-intensive, this inevitably increases training and inference time.

Please refer to Sec. D.1 and Sec. F.2 for the exact numbers of training time on QM9 and OC20.

# D  DETAILS OF EXPERIMENTS ON QM9

## D.1  TRAINING DETAILS

We use the same data partition as TorchMD-NET (Thölke & Fabritiis, 2022). For the task of $U$, $U_0$, $G$, and $H$, where single-atom reference values are available, we subtract those reference values from ground truth. For other tasks, we normalize ground truth by subtracting mean and dividing by standard deviation.

We train Equiformer with 6 blocks with $L_{max} = 2$. We choose Gaussian radial basis (Schütt et al., 2017; Shuaibi et al., 2021; Klicpera et al., 2021; Shi et al., 2022) for the first six tasks in Table 1 and radial Bessel basis (Gasteiger et al., 2020b;a) for the others. We apply dropout (Srivastava et al., 2014) to attention weights $a_{ij}$. The dropout rate is 0.0 for the tasks of $G$, $H$, $U$ and $U_0$, is 0.1 for the task of $R^2$ and is 0.2 for others. Since the tasks of $G$, $H$, $U$, and $U_0$ require longer training, we use slightly different hyper-parameters. The number of epochs is 600 for the tasks of $G$, $H$, $U$, and $U_0$ and is 300 for others. The learning rate is $1.5 \times 10^{-4}$ for the tasks of $G$, $H$, $U$, and $U_0$ and is $5 \times 10^{-4}$ for others. The batch size is 64 for the tasks of $G$, $H$, $U$, and $U_0$ and is 128 for others. The weight decay is 0 for the tasks of $G$, $H$, $U$, and $U_0$ and is $5 \times 10^{-3}$ for others. Table 8 summarizes the hyper-parameters for the QM9 dataset. The detailed description of architectural hyper-parameters can be found in Sec. C.2.

We use one A6000 GPU with 48GB to train each model and summarize the computational cost of training for one epoch as follows. Training $E(3)$-Equiformer in Table 9 for one epoch takes about 16.3 minutes. The time of training Equiformer, Equiformer with linear messages (indicated by Index 2 in Table 6), and Equiformer with linear messages and dot product attention (indicated by Index 3 in Table 6) for one epoch is 12.1 minutes, 7.2 minutes and 7.8 minutes, respectively.

## D.2    COMPARISON BETWEEN $SE(3)$ AND $E(3)$ EQUIVARIANCE

We train two versions of Equiformers, one with $SE(3)$-equivariant features denoted as "Equiformer" and the other with $E(3)$-equivariant features denoted as "$E(3)$-Equiformer", and we compare them in Table 9. As for Table 1, we compare "Equiformer" with other works since most of them do not include equivariance to inversion.

| Methods | Task Units | $\alpha$ $a_0^3$ | $\Delta\varepsilon$ meV | $\varepsilon_{\text{HOMO}}$ meV | $\varepsilon_{\text{LUMO}}$ meV | $\mu$ D | $C_\nu$ cal/mol K | Training time (minutes/epoch) | Number of parameters |
|---------|-----------|------|-----|------|------|-----|------|------------|-----------|
| Equiformer | | .046 | 30 | 15 | 14 | .011 | .023 | 12.1 | 3.53M |
| $E(3)$-Equiformer | | .045 | 30 | 15 | 14 | .012 | .023 | 16.3 | 3.28M |

Table 9: **Ablation study of $SE(3)$/$E(3)$ equivariance on QM9 testing set.** "Equiformer" operates on $SE(3)$-equivariant features while "$E(3)$-Equiformer" uses $E(3)$-equivariant features. Including inversion achieves similar performance.

## D.3    COMPARISON OF TRAINING TIME AND NUMBERS OF PARAMETERS

We compare training time and numbers of parameters between SEGNN (Brandstetter et al., 2022), TorchMD-NET (Thölke & Fabritiis, 2022) and Equiformer and summarize the results in Table 10. Training Equiformer for 300 epochs and for 600 epochs takes 61 and 122 GPU-hours, respectively.

Compared to SEGNN, which is written with the same e3nn library (Geiger et al., 2022), Equiformer with MLP attention and non-linear message is faster. Although Equiformer has more channels and more parameters, the training time is comparable. The reasons are as follows. Equiformer uses more efficient depth-wise tensor products (DTP), where one output channel depends on only one input channel. SEGNN uses more compute-intensive fully connected tensor products (FCTP), where one output channel depends on all input channels. Besides, SEGNN uses 4 FCTPs in each message passing block while Equiformer uses only 2 DTPs in each block.

Compared to TorchMD-NET, which is trained for 3000 epochs, Equiformer achieves competitve results after trained for 300 or 600 epochs. Equiformer takes more time for each epoch since Equiformer uses more expressive non-linear messages, which compared to linear messages used in other equivariant Transformers, doubles the number of tensor products and therefore almost doubles the training time. Moreover, Equiformer incorporates tensors of higher degrees (e.g., $L_{max} = 2$), which improves performance but slows down the training.

# E    DETAILS OF EXPERIMENTS ON MD17

## E.1    TRAINING DETAILS

We use the same data partition as TorchMD-NET (Thölke & Fabritiis, 2022). For energy prediction, we normalize ground truth by subtracting mean and dividing by standard deviation. For force prediction, we normalize ground truth by dividing by standard deviation of ground truth energy.

We train Equiformer with 6 blocks with $L_{max} = 2$ and 3. We choose the radial basis function used by PhysNet (Unke & Meuwly, 2019). We do not apply dropout to attention weights $a_{ij}$. For Equiformer with $L_{max} = 2$, the learning rate is $1 \times 10^{-4}$ for benzene and is $5 \times 10^{-4}$ for others. The batch size is 8, and the number of epochs is 1500. The model has about 3.50M parameters. For Equiformer with $L_{max} = 3$, the learning rate is $1 \times 10^{-4}$ for benzene and is $2 \times 10^{-4}$ for others. The batch size is 5, and the number of epochs is 2000. The model has about 5.50M parameters. Table 11 summarizes the hyper-parameters for the MD17 dataset. The detailed description of architectural hyper-parameters can be found in Sec. C.2.

| Methods | Number of parameters | Training time (GPU-hours) |
|---|---|---|
| SEGNN (Brandstetter et al., 2022) | 1.03M | 81 |
| TorchMD-NET (Thölke & Fabritiis, 2022) | 6.86M | 92 |
| Equiformer | 3.53M | 61 |

Table 10: **Comparison of training time and numbers of parameters for QM9 dataset.**

| Hyper-parameters | Value or description |
|---|---|
| Optimizer | AdamW |
| Learning rate scheduling | Cosine learning rate with linear warmup |
| Warmup epochs | 10 |
| Maximum learning rate | $1 \times 10^{-4}, 2 \times 10^{-4}, 5 \times 10^{-4}$ |
| Batch size | 5, 8 |
| Number of epochs | 1500, 2000 |
| Weight decay | $1 \times 10^{-6}$ |
| Dropout rate | 0.0 |
| Weight for energy loss | 1 |
| Weight for force loss | 80 |
| Cutoff radius (Å) | 5 |
| Number of radial bases | 32 |
| Hidden sizes of radial functions | 64 |
| Number of hidden layers in radial functions | 2 |
| Equiformer ($L_{max} = 2$) | |
| Number of Transformer blocks | 6 |
| Embedding dimension $d_{embed}$ | $[(128, 0), (64, 1), (32, 2)]$ |
| Spherical harmonics embedding dimension $d_{sh}$ | $[(1, 0), (1, 1), (1, 2)]$ |
| Number of attention heads $h$ | 4 |
| Attention head dimension $d_{head}$ | $[(32, 0), (16, 1), (8, 2)]$ |
| Hidden dimension in feed forward networks $d_{ffn}$ | $[(384, 0), (192, 1), (96, 2)]$ |
| Output feature dimension $d_{feature}$ | $[(512, 0)]$ |
| Equiformer ($L_{max} = 3$) | |
| Number of Transformer blocks | 6 |
| Embedding dimension $d_{embed}$ | $[(128, 0), (64, 1), (64, 2), (32, 3)]$ |
| Spherical harmonics embedding dimension $d_{sh}$ | $[(1, 0), (1, 1), (1, 2), (1, 3)]$ |
| Number of attention heads $h$ | 4 |
| Attention head dimension $d_{head}$ | $[(32, 0), (16, 1), (16, 2), (8, 3)]$ |
| Hidden dimension in feed forward networks $d_{ffn}$ | $[(384, 0), (192, 1), (192, 2), (96, 3)]$ |
| Output feature dimension $d_{feature}$ | $[(512, 0)]$ |

Table 11: **Hyper-parameters for MD17 dataset.** We denote $C_L$ type-$L$ vectors as $(C_L, L)$ and $C_{(L,p)}$ type-$(L, p)$ vectors as $(C_{(L,p)}, L, p)$ and use brackets to represent concatenations of vectors.

We use one A5000 GPU with 24GB to train different models for each molecule. Training Equiformer with $L_{max} = 2$ takes about 15.4 hours, and training Equiformer with $L_{max} = 3$ takes about 54.4 hours.

### E.2   ADDITIONAL COMPARISON TO TORCHMD-NET

Since TorchMD-NET (Thölke & Fabritiis, 2022) is also an equivariant Transformer but uses dot product attention instead of the proposed equivariant graph attention, we provide additional comparisons in Table 12. For each molecule, we adjust the ratio of the weight for force loss to the weight for energy loss so that Equiformer with $L_{max} = 2$ can achieve lower MAE for both energy and forces.

### E.3   COMPARISON OF TRAINING TIME AND NUMBERS OF PARAMETERS

We compare training time and number of parameters between NequIP (Batzner et al., 2022) and Equiformer and summarize the results in Table 13. Since NequIP does not report the number of epochs, we compare the time spent for each epoch and note that NequIP is trained for more than 1000 epochs. Equiformer with $L_{max} = 2$ is faster than NequIP with $L_{max} = 3$ since smaller $L_{max}$ is used. Moreover, Equiformer with $L_{max} = 2$ achieves overall better results as we use equivariant graph attention instead of linear convolution used by NequIP. When increasing $L_{max}$ from 2 to 3,

| Methods | Aspirin | | Benzene | | Ethanol | | Malonaldehyde | | Naphthalene | | Salicylic acid | | Toluene | | Uracil | |
|---|---|---|---|---|---|---|---|---|---|---|---|---|---|---|---|---|
| | energy | forces | energy | forces | energy | forces | energy | forces | energy | forces | energy | forces | energy | forces | energy | forces |
| TorchMD-NET | **5.3** | 11.0 | 2.5 | 8.5 | 2.3 | 4.7 | 3.3 | 7.3 | **3.7** | 2.6 | **4.0** | 5.6 | **3.2** | 2.9 | **4.1** | 4.1 |
| Equiformer ($L_{max} = 2$) | **5.3** | 7.2 | **2.2** | 6.6 | **2.2** | 3.1 | 3.3 | 5.8 | **3.7** | 2.1 | **4.0** | 5.3 | **3.2** | 2.4 | 4.2 | **3.7** |

Table 12: **Additional comparison to TorchMD-Net (Thölke & Fabritiis, 2022) on MD17 dataset.** Energy and force are in units of meV and meV/Å.

| Methods | Number of parameters | Training time (secs/epoch) |
|---|---|---|
| NequIP ($L_{max} = 3$) (Batzner et al., 2022) | 2.97M | 48.7 |
| Equiformer ($L_{max} = 2$) | 3.50M | 36.9 |
| Equiformer ($L_{max} = 3$) | 5.50M | 98.0 |

Table 13: **Comparison of training time and numbers of parameters for MD17 dataset.**

Equiformer achieves better results than NequIP for most molecules. However, the training time is longer than NequIP since we use 2 tensor product operations in each equivariant graph attention instead of 1 tensor product in each linear convolution.

# F    DETAILS OF EXPERIMENTS ON OC20

## F.1    DETAILED DESCRIPTION OF OC20 DATASET

The dataset consists of larger atomic systems, each composed of a molecule called adsorbate placed on a slab called catalyst. Each input contains more atoms and more diverse atom types than QM9 and MD17. The average number of atoms in a system is more than 70, and there are over 50 atom species. The goal is to understand interaction between adsorbates and catalysts through relaxation. An adsorbate is first placed on top of a catalyst to form initial structure (IS). The positions of atoms are updated with forces calculated by density function theory until the system is stable and becomes relaxed structure (RS). The energy of RS, or relaxed energy (RE), is correlated with catalyst activity and therefore a metric for understanding their interaction. We focus on the task of initial structure to relaxed energy (IS2RE), which predicts relaxed energy (RE) given an initial structure (IS). There are 460k, 100k and 100k structures in training, validation, and testing sets, respectively.

## F.2    TRAINING DETAILS

**IS2RE without Node-Level Auxiliary Task.**    We use hyper-parameters similar to those for QM9 dataset and summarize in Table 14. For ablation study in Sec. 5.4, we use a smaller learning rate $1.5 \times 10^{-4}$ for DP attention as this improves the performance. The detailed description of architectural hyper-parameters can be found in Sec. C.2.

**IS2RE with IS2RS Node-Level Auxiliary Task.**    We increase the number of Transformer blocks to 18 as deeper networks can benefit more from IS2RS node-level auxiliary task (Godwin et al., 2022). We follow the same hyper-parameters in Table 14 except that we increase maximum learning rate to $5 \times 10^{-4}$ and set $d_{feature}$ to $[(512, 0), (256, 1)]$. Inspired by Graphormer (Shi et al., 2022), we add an extra equivariant graph attention module after the last layer normalization to predict relaxed structures and use a linearly decayed weight for loss associated with IS2RS, which starts at 15 and decays to 1. For Noisy Nodes (Godwin et al., 2022) data augmentation, we first interpolate between initial structure and relaxed structure and then add Gaussian noise as described by Noisy Nodes (Godwin et al., 2022). When Noisy Nodes data augmentation is used, we increase the number of epochs to 40.

We use two A6000 GPUs, each with 48GB, to train models when IS2RS is not included during training. Training Equiformer and $E(3)$-Equiformer in Table 16 takes about 43.6 and 58.3 hours. Training Equiformer with linear messages (indicated by Index 2 in Table 7) and Equiformer with linear messages and dot product attention (indicated by Index 3 in Table 7) takes 30.4 hours and 33.1 hours, respectively. We use four A6000 GPUs to train Equiformer models when IS2RS node-level auxiliary task is adopted during training. Training Equiformer without Noisy Nodes data augmentation takes about 3 days and training with Noisy Nodes takes 6 days. We note that the proposed Equiformer in Table 5 achieves competitive results even with much less computation. Specifically, training

| Hyper-parameters | Value or description |
|---|---|
| Optimizer | AdamW |
| Learning rate scheduling | Cosine learning rate with linear warmup |
| Warmup epochs | 2 |
| Maximum learning rate | $2 \times 10^{-4}$ |
| Batch size | 32 |
| Number of epochs | 20 |
| Weight decay | $1 \times 10^{-3}$ |
| Dropout rate | 0.2 |
| Cutoff radius (Å) | 5 |
| Number of radial basis | 128 |
| Hidden size of radial function | 64 |
| Number of hidden layers in radial function | 2 |
| *Equiformer* | |
| Number of Transformer blocks | 6 |
| Embedding dimension $d_{embed}$ | $[(256, 0), (128, 1)]$ |
| Spherical harmonics embedding dimension $d_{sh}$ | $[(1, 0), (1, 1)]$ |
| Number of attention heads $h$ | 8 |
| Attention head dimension $d_{head}$ | $[(32, 0), (16, 1)]$ |
| Hidden dimension in feed forward networks $d_{ffn}$ | $[(768, 0), (384, 1)]$ |
| Output feature dimension $d_{feature}$ | $[(512, 0)]$ |
| *$E(3)$-Equiformer* | |
| Number of Transformer blocks | 6 |
| Embedding dimension $d_{embed}$ | $[(256, 0, e), (64, 0, o), (64, 1, e), (64, 1, o)]$ |
| Spherical harmonics embedding dimension $d_{sh}$ | $[(1, 0, e), (1, 1, o)]$ |
| Number of attention heads $h$ | 8 |
| Attention head dimension $d_{head}$ | $[(32, 0, e), (8, 0, o), (8, 1, e), (8, 1, o)]$ |
| Hidden dimension in feed forward networks $d_{ffn}$ | $[(768, 0, e), (192, 0, o), (192, 1, e), (192, 1, o)]$ |
| Output feature dimension $d_{feature}$ | $[(512, 0, e)]$ |

Table 14: **Hyper-parameters for OC20 dataset under the setting of training without IS2RS auxiliary task.** We denote $C_L$ type-$L$ vectors as $(C_L, L)$ and $C_{(L,p)}$ type-$(L, p)$ vectors as $(C_{(L,p)}, L, p)$ and use brackets to represent concatenations of vectors.

| Methods | Energy MAE (eV) ↓ | | | | | EwT (%) ↑ | | | | |
|---|---|---|---|---|---|---|---|---|---|---|
| | ID | OOD Ads | OOD Cat | OOD Both | Average | ID | OOD Ads | OOD Cat | OOD Both | Average |
| SchNet (Schütt et al., 2017)[†] | 0.6465 | 0.7074 | 0.6475 | 0.6626 | 0.6660 | 2.96 | 2.22 | 3.03 | 2.38 | 2.65 |
| DimeNet++ (Gasteiger et al., 2020a)[†] | 0.5636 | 0.7127 | 0.5612 | 0.6492 | 0.6217 | 4.25 | 2.48 | 4.40 | 2.56 | 3.42 |
| GemNet-T (Klicpera et al., 2021)[†] | 0.5561 | 0.7342 | 0.5659 | 0.6964 | 0.6382 | 4.51 | 2.24 | 4.37 | 2.38 | 3.38 |
| SphereNet (Liu et al., 2022) | 0.5632 | 0.6682 | 0.5590 | 0.6190 | 0.6024 | 4.56 | 2.70 | 4.59 | 2.70 | 3.64 |
| (S)EGNN (Brandstetter et al., 2022) | 0.5497 | 0.6851 | 0.5519 | 0.6102 | 0.5992 | 4.99 | 2.50 | 4.71 | 2.88 | 3.77 |
| SEGNN (Brandstetter et al., 2022) | 0.5310 | 0.6432 | 0.5341 | 0.5777 | 0.5715 | **5.32** | 2.80 | 4.89 | **3.09** | **4.03** |
| Equiformer | **0.5088** | **0.6271** | **0.5051** | **0.5545** | **0.5489** | 4.88 | **2.93** | **4.92** | 2.98 | 3.93 |

Table 15: **Results on OC20 IS2RE validation set.** † denotes results reported by Liu et al. (2022).

"Equiformer + Noisy Nodes" takes about 24 GPU-days when A6000 GPUs are used. The training time of "GNS + Noisy Nodes" (Godwin et al., 2022) is 56 TPU-days. "Graphormer" (Shi et al., 2022) uses ensemble of 31 models and requires 372 GPU-days to train all models when A100 GPUs are used.

### F.3 RESULTS ON IS2RE VALIDATION SET

For completeness, we report the results of Equiformer on the validation set in Table 15.

### F.4 COMPARISON BETWEEN $SE(3)$ AND $E(3)$ EQUIVARIANCE

We train two versions of Equiformers, one with $SE(3)$-equivariant features denoted as "Equiformer" and the other with $E(3)$-equivariant features denoted as "$E(3)$-Equiformer", and we compare them in Table 16. Including inversion improves the MAE results on ID and OOD Cat sub-splits but degrades the performance on the other sub-splits. Overall, using $E(3)$-equivariant features results in slightly inferior performance. We surmise the reasons are as follows. First, inversion might not be the key bottleneck. Second, including inversion would break type-1 vectors into two parts, type-$(1, e)$

| Methods | Energy MAE (eV) ↓ | | | | | EwT (%) ↑ | | | | | Training time (minutes/epoch) | Number of parameters |
|---|---|---|---|---|---|---|---|---|---|---|---|---|
| | ID | OOD Ads | OOD Cat | OOD Both | Average | ID | OOD Ads | OOD Cat | OOD Both | Average | | |
| Equiformer | 0.5088 | 0.6271 | 0.5051 | 0.5545 | 0.5489 | 4.88 | 2.93 | 4.92 | 2.98 | 3.93 | 130.8 | 9.12M |
| $E(3)$-Equiformer | 0.5035 | 0.6385 | 0.5034 | 0.5658 | 0.5528 | 5.10 | 2.98 | 5.10 | 3.02 | 4.05 | 174.9 | 8.77M |

Table 16: **Ablation study of $SE(3)/E(3)$ equivariance on OC20 IS2RE validation set.** "Equiformer" operates on $SE(3)$-equivariant features while "$E(3)$-Equiformer" uses $E(3)$-equivariant features.

| Methods | Number of parameters | Training time (GPU-hours) |
|---|---|---|
| SEGNN (Brandstetter et al., 2022) | 4.21M | 79 |
| Equiformer | 9.12M | 87 |

Table 17: **Comparison of training time and numbers of parameters for OC20 dataset.**

and type-$(1, o)$ vectors. They are regarded as different types in equivariant linear layers and layer normalizations, and therefore, the directional information captured in these two types of vectors can only exchange in depth-wise tensor products. Third, we mainly tune hyper-parameters for Equiformer with $SE(3)$-equivariant features, and it is possible that using $E(3)$-equivariant features would favor different hyper-parameters.

For Table 15, 3, 4, and 5, we compare "Equiformer" with other works since most of them do not include equivariance to inversion.

### F.5 COMPARISON OF TRAINING TIME AND NUMBERS OF PARAMETERS

We compare training time and numbers of parameters between SEGNN (Brandstetter et al., 2022) and Equiformer when IS2RS auxiliary task is not adopted during training and summarize the results in Table 17. Equiformer achieves better results with comparable training time. Please refer to Sec. D.3 for a detailed discussion.

The comparison of training time when IS2RS auxiliary task is adopted can be found in Table 5.

### F.6 ERROR DISTRIBUTIONS

We plot the error distributions of different Equiformer models on different sub-splits of OC20 IS2RE validation set in Fig. 5. For each curve, we sort the absolute errors in ascending order for better visualization and have a few observations. First, for each sub-split, there are always easy examples, for which all models achieve significantly low errors, and hard examples, for which all models have high errors. Second, the performance gains brought by different models are non-uniform among different sub-splits. For example, using MLP attention and non-linear messages improves the errors on the ID sub-split but is not that helpful on the OOD Ads sub-split. Third, when IS2RS node-level auxiliary task is not included during training, using stronger models mainly improves errors that are beyond the threshold of 0.02 eV, which is used to calculate the metric of energy within threshold (EwT). For instance, on the OOD Both sub-split, using non-linear messages, which corresponds to red and purple curves, improves the absolute errors for the 15000th through 20000th examples. However, the improvement in MAE does not translate to that in EwT as the errors are still higher than the threshold of 0.02 eV. This explains why using non-linear messages in Table 7 improves MAE from 0.5657 to 0.5545 but results in almost the same EwT.

## G LIMITATIONS

We discuss several limitations of the proposed Equiformer and equivariant graph attention below.

First, Equiformer is based on irreducible representations (irreps) and therefore can inherit the limitations common to all equivariant networks based on irreps and the library e3nn (Geiger et al., 2022). For example, using higher degrees $L$ can result in larger features and using tensor products can be compute-intensive. Part of the reasons that tensor products can be computationally expensive are that the kernels have not been heavily optimized and customized as other operations in common libraries like PyTorch (Paszke et al., 2019). But this is the issue related to software, not the design of

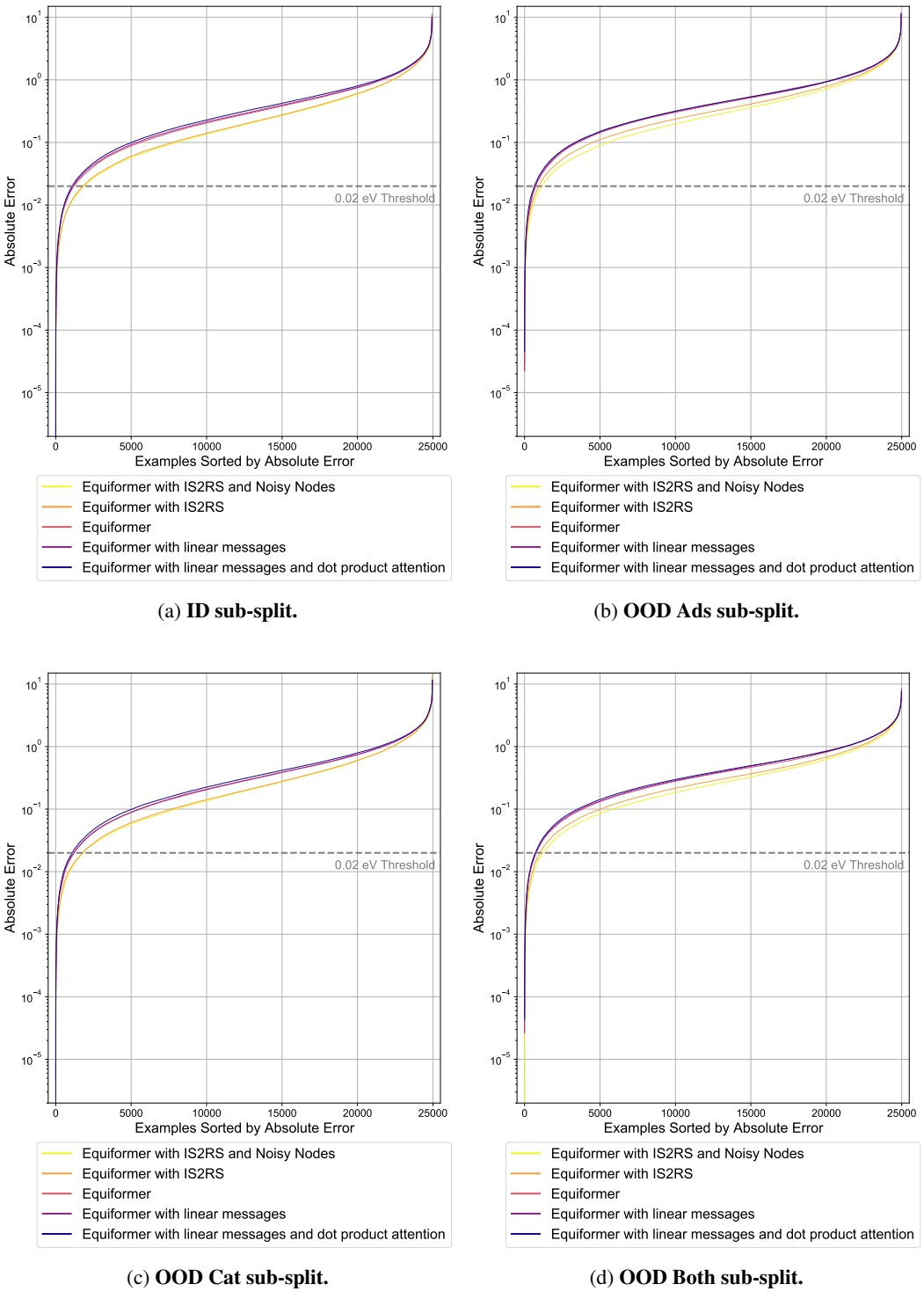

Figure 5: **Error distributions of different Equiformer models on different sub-splits of OC20 IS2RE validation set.**

networks. While tensor products of irreps naively do not scale well, if all possible interactions and paths are considered, some paths in tensor products can also be pruned for computational efficiency. We leave these potential efficiency gains to future work and in this work focus on general equivariant attention if all possible paths up to $L_{max}$ in tensor products are allowed.

Second, the improvement of the proposed equivariant graph attention can depend on tasks and datasets. For QM9, MLP attention improves not significantly upon dot product attention as shown in Table 6. We surmise that this is because QM9 contains less atoms and less diverse atom types and therefore linear attention is enough. For OC20, MLP attention clearly improves upon dot product attention as shown in Table 7. Non-linear messages improve upon linear ones for the two datasets.

Third, equivariant graph attention requires more computation than typical graph convolution. It includes one softmax operation and thus requires one additional sum aggregation compared to typical message passing. For non-linear message passing, it increases the number of tensor products from one to two and requires more computation. We note that if there is a constraint on training budget, using stronger attention (i.e., MLP attention and non-linear messages) would not always be optimal because for some tasks or datasets, the improvement is not that significant and using stronger attention can slow down training.

Fourth, the proposed attention has complexity proportional to the products of numbers of channels and numbers of edges since the the attention is restricted to local neighborhoods. In the context of 3D atomistic graphs, the complexity is the same as that of messages and graph convolutions. However, in other domains like computer vision, the memory complexity of convolution is proportional to the number of pixels or nodes, not that of edges. Therefore, it would require further modifications in order to use the proposed attention in other domains.

