# OpenReview forum: "Equiformer: Equivariant Graph Attention Transformer for 3D Atomistic Graphs"
_ICLR.cc/2023/Conference — ICLR 2023 notable top 25%_

### Official Review · Reviewer_8jGX · 2022-10-23

**Confidence:** 2
**Correctness:** 3
**Technical Novelty And Significance:** 3
**Empirical Novelty And Significance:** 2
**Recommendation:** 5

**Clarity, Quality, Novelty And Reproducibility:**

I'm a novice at the equivariance or the group theory. Therefore, the overall impression on this paper is "difficult to read", The detailed appendix sections (thank you, authors!!) helped me a lot to understand, but I still need some questions to (better) understand the contributions of this paper.

Please feel free to correct my misunderstandings!

[main contributions]
Consists of the following two items. Is this correct?
* A Transformer that can handle any equivalent features (more than type-0 and type-1 vectors)
* Developed an equivalent operator for a MLP-driven attention with nonlinear activations

[technical novelty]
* All the contents of Sec 4.1 are basically established in the existing literature?
* I understand Sec 4.2 is the core of the technical novelty, especially Eqs. (2-4). Please give me some more explanations about (I) where the difficulty lies, (ii) how difficult the challenges are, and (iii) how you tackle the challenges.
* Nonlinear equivariant messaging (Eq.5) is a known result (e.g. by SEGNN)?

[some naive questions]
* Please consider presenting some simple examples. For example, how the type-L vectors look like for each symmetry?
* Fig. 2 What are the meanings of the colors of nodes?



**Strength And Weaknesses:**

(+) Developed a generic MLP attention operator with a non-linear activation function.


(+) Experimental results show the efficacy of the proposed method against the existing equivariant models.

(-) difficult to read especially for readers who are unfamiliar with equivariance or group theory.

(-) As a result, the technical novelty of the proposed method is unclear (to me, at least)

(-) The difficulty of the tackled technical challenge is not discussed. This makes it difficult for me to understand the significance of the proposed solution in the equivariant GNN literature.

**Summary Of The Paper:**

An equivariant Transformer/GNN model is proposed to deal with the 3D symmetries (rotation, transliteration, and inversion). The main targets are atomic graphs such as molecules.
The main technical advancement of this paper is the equivariant attention operation with multi-layer perception and nonlinear activations.
Experimentally, the proposed model records better figures consistently compared to the existing equivariant-aware Transformer/GNNs.

**Summary Of The Review:**

A Transformer with 3D equivariant awareness is presented. The main contribution is the MLP-based attention that is capable of equivariant outcomes.
Experimental results are strong.
The manuscript is not easy to read for readers unfamiliar with equivariant topics. More explanations through feedback is expected to more correctly evaluate the paper.

---

> ### Author Response · Authors · 2022-11-15
> **Response to Reviewer 8jGX (1/4)**
>
> We thank the reviewer for helpful feedback and address the comments below.
>
> ---
> > 1. Difficult to read especially for readers who are unfamiliar with equivariance or group theory.
>
> We respectfully disagree that this paper is generally difficult to read. As mentioned by the reviewer **sfAh**, the paper is well-written, and we have discussed the differences between previous works and this work in detail and made clear the proposed architecture.  Even though it can be difficult for people not familiar with equivariance and group theory, we have already provided sufficient background in Section 3 and Section A. Besides, we also give some relevant intuition like why it is important to include equivariance in neural networks in Section A. We note that it is impossible to include all background of group theory in a technical paper, but we have pointed out useful resources in Section A.
>
> Generally, it is unclear which part of the presentation needs to be improved and it can be very helpful if the reviewer can point out which part is difficult to read.
>
> ---
> > 2. The technical novelty of the proposed method is unclear.
>
> Please see **General Response 1** for the contribution of Equiformer compared to other equivariant Transformers, **General Response 2** for novelties of this work and **General Response 3** for detailed comparison between equivariant Transformers.

---

> > ### Author Response · Authors · 2022-11-15
> > **Response to Reviewer 8jGX (2/4)**
> >
> > > 3. The difficulty of the tackled technical challenge is not discussed, which makes it difficult to understand the significance of the proposed solution in the equivariant GNN literature. Please give me some more explanations about (i) where the difficulty lies, (ii) how difficult the challenges are, and (iii) how you tackle the challenges.
> >
> > Generally speaking, we investigate whether Transformers can be as powerful models in the domain of learning representations of 3D atomistic graphs as they have been in other domains and then develop an effective architecture. We discuss some relevant challenges below.
> >
> > (i) Where do the difficulties lie?
> > ---
> >
> > **a. Invariant or equivariant representations?**
> > For the domain of 3D atomistic graphs, there are two different types of networks, invariant and equivariant networks. They differ in the internal representations and the operations included in their networks, and choosing which of them can significantly affect which operations we can use when designing networks.
> >
> >
> > **b. How do Transformers generalize to other domains?**
> > Since previous works on equivariant Transformers have not achieved promising results across a wide range of datasets in the domain of 3D atomistic graphs, it is unclear whether Transformers can generalize to this domain and we should learn how Transformers generalize to other domains.
> >
> > **c. What are the minimal modifications required to obtain an effective Transformer for 3D atomistic graphs?**
> > One motivation to use Transformers for various domains is to see how this simple architecture can generalize. If the architecture is simpler, it is more likely to be used by other researchers and adapt to other problems. For example, Equiformer has been used for protein structure prediction in the work of EquiFold [1].
> >
> > **d. What are other modifications that are not present in Transformers in other domains but we can have for 3D atomistic graphs?**
> >
> >
> > (ii) How difficult are the challenges?
> > ---
> >
> > **a.** This requires understanding the details, advantages and disadvantages of both invariant and equivariant networks as well as some theories behind them (e.g., group theory).
> >
> > **b.** This requires understanding specific structure of data in various domains like computer vision and graphs and how Transformers incorporate domain-specific knowledge into networks.
> >
> > **c.** This requires trying some architecture variants commonly used in other message passing networks to see whether the architecture of Transformers can be improved in this domain. For example, message passing networks typically concatenate node features and edge features and then transform the joint features.
> >
> > **d.** This requires understanding modifications researchers have done for attention in other domains so that we can get inspiration. Additionally, this requires understanding the computational cost of each component in equivariant networks.
> >
> > (iii) How to tackle the challenges?
> > ---
> >
> > **a.** We decide to use equivariant networks since they can encode geometric information without complicating graph structures as mentioned in the sections of abstract and introduction. This decision makes processing 3D atomistic graph data more similar to processing image or language data compared to using invariant networks and enables direct applications of generic GNNs.
> >
> > **b.** The strong inductive bias of 3D atomistic graphs is E(3) equivariance. Since we already choose equivariant networks, the internal representations incorporate the inductive bias.
> >
> > **c.** We empirically find that those variants provide almost no performance gain, and the original Transformer architecture is sufficient. The only difference lies in we include additional tensor product operations, which is to mix information contained in vectors of different degrees L and is well-justified.
> >
> > **d.** We observe that generally the messages or features sent from one node to another can be decomposed into attention weights and value vectors as in Equation (2). This observation motivates using more expressive functions for both attention weights and value vectors to achieve overall better expressiveness. Since non-linear value vectors have the same computation and memory complexity as typical message passing in equivariant networks, this design is feasible. Note that the combination of MLP attention and non-linear messages cannot be trivially applied to domains like computer vision and natural language processing due to the memory complexity as discussed in Section G.
> >
> > Reference:
> >
> > [1] Lee et al. EquiFold: Protein Structure Prediction with a Novel
> > Coarse-Grained Structure Representation. bioRxiv 2022.

---

> > > ### Author Response · Authors · 2022-11-15
> > > **Response to Reviewer 8jGX (3/4)**
> > >
> > > > 4. [main contributions] Consists of the following two items. Is this correct? (a) A Transformer that can handle any equivalent features (more than type-0 and type-1 vectors). (b) Developed an equivalent operator for a MLP-driven attention with nonlinear activations.
> > >
> > > Yes.
> > >
> > > ---
> > >
> > > > 5. All the contents of Sec 4.1 are basically established in the existing literature?
> > >
> > > Yes, they either directly come from previous works or modified from them. Please see **General Response 4** for further details.
> > >
> > > ---
> > > > 6. Nonlinear equivariant messaging (Eq.5) is a known result (e.g. by SEGNN)?
> > >
> > > Yes. To be more specific, the idea of non-linear message passing has been proposed and used by other works [1, 2, 3] before SEGNN [4]. The contribution of SEGNN lies in developing the equivariant architecture of non-linear message passing by using gate activation [5] and demonstrating this works well for QM9 and OC20.
> > >
> > > In contrast, the idea of combining MLP attention and non-linear message passing in our proposed manner has not been explored in previous works and neither has the corresponding equivariant version. Moreover, we show that this new combination works for a wider range of datasets (MD17, QM9 and OC20).
> > >
> > > Reference:
> > >
> > > [1] Gilmer et al. Neural message passing for quantum chemistry. ICML 2017.
> > >
> > > [2] Xie et al. Crystal Graph Convolutional Neural Networks for an Accurate and Interpretable Prediction of Material Properties. Physical Review Letters 2018.
> > >
> > > [3] Hu et al. ForceNet: A Graph Neural Network for Large-Scale Quantum Calculations. ICLR 2021 Workshop.
> > >
> > > [4] Brandstetter et al. Geometric and Physical Quantities improve E(3) Equivariant Message Passing. ICLR 2022 Spotlight.
> > >
> > > [5] Weiler et al.  3D Steerable CNNs: Learning Rotationally Equivariant Features in Volumetric Data. NeurIPS 2018.
> > >
> > > ---
> > > > 7. Please consider presenting some simple examples. For example, what do the type-L vectors look like for each symmetry?
> > >
> > > Please see the paragraph of “Vectors of Higher L and Other Parities” in Section A.3 and the paragraph of “Examples of Tensor Products” in Section A.4 for some examples of type-L vectors. Additionally, the visualization of spherical harmonics can be found in the website [1], and we will add this to the appendix.
> > >
> > > Please let us know if these examples are sufficient.
> > >
> > > [1] https://docs.e3nn.org/en/latest/api/o3/o3_sh.html

---

> > > > ### Author Response · Authors · 2022-11-15
> > > > **Response to Reviewer 8jGX (4/4)**
> > > >
> > > > > 8. Fig. 2 What are the meanings of the colors of nodes?
> > > >
> > > > As already shown in the figure, different colors correspond to vectors of different degree L. Take Figure 2(a) for example. The pink nodes correspond to type-0 vectors (L = 0). Since the dimension of each type-0 vector is one, each instance of type-0 vector contains only one square. The number of channels for type-0 vectors is four, so there are four instances of type-0 vectors and totally four pink nodes ($4$ (number of channels) $\times$ $1$ (dimension of each type-0 vector)). Similarly, for type-1 vectors (green nodes), the dimension of each type-1 vector is three and the number of channels for type-1 vectors is three. For type-2 vectors (blue nodes), the dimension of each type-2 vector is five and the number of channels for type-2 vectors is two.

---

> > > > > ### Comment · Reviewer_8jGX · 2022-11-16
> > > > > **Thanks authors!**
> > > > >
> > > > > Thank you for your detailed answers. These will help me understand the value of the manuscript.
> > > > > Especially, I find the answer section 2(/n) is highly helpful.
> > > > >
> > > > > I will check the updated manuscript and your feedbacks. Thank you again!

---

> ### Author Response · Authors · 2022-12-07
> **Please let us know if you have other questions or comments**
>
> We believe we have addressed your concerns and provided detailed descriptions of relevant challenges and contributions.
>
> Please let us know if you have any feedback to our explanation or other questions. We will be very happy to respond.

---

### Official Review · Reviewer_1yMo · 2022-10-25

**Confidence:** 4
**Correctness:** 3
**Technical Novelty And Significance:** 2
**Empirical Novelty And Significance:** 3
**Recommendation:** 6

**Clarity, Quality, Novelty And Reproducibility:**

The writing quality is pool. Sec. 4 just introduce each component one-by-one. The authors should point out their own contributions in the paper and why the proposed method is able to keep the equivariance.

**Strength And Weaknesses:**

Strength:
1. The proposed method considers various equivariances in transformer frameworks.
2. The experimental results of IS2RE for OC20 dataset outperforms the other baselines.

Weakness:
1. In Sec. 4, all the components of equiformer are existing modules. The authors should emphasis their contributions or modifications to these existing modules.
2. In Sec. 4, the analyses of equivariance are missing. Why the proposed components still keep equivariance?
3. It seems that the performances on MD17 and QM9 are marginal. Why the baselines evaluated on MD17, QM9, and OC20 are different?


**Summary Of The Paper:**

The paper proposes a novel transformer framework that involve equivariance for the inputs. It shows high performance on OC20 dataset and comparable results on MD17 and QM9.

**Summary Of The Review:**

Overall, the novelty, experimental evaluations, and the motivation are enough in the current form. However, the paper is hard to follow because the novel part of the proposed method and the existing modules are mixed in the paper. The writing should be improved

---

> ### Author Response · Authors · 2022-11-15
> **Response to Reviewer 1yMo (1/2)**
>
> We thank the reviewer for helpful feedback and address the comments below.
>
> ---
> > 1. In Sec. 4, all the components of Equiformer are existing modules. The authors should emphasize their contributions or modifications to these existing modules.
>
> Please see **General Response 1** for the contribution of Equiformer compared to other equivariant Transformers, **General Response 2** for novelties of this work and **General Response 3** for detailed comparison between equivariant Transformers.
>
> ---
> > 2. The paper is hard to follow because the novel part of the proposed method and the existing modules are mixed in the paper.
>
> Please see **General Response 4** for detailed description of which part is novel and which part is previously proposed.
>
> We justify the writing below. In Section 4.1, we figure out the four minimal equivariant operations that we can use to build the whole network. Some operations are directly borrowed from previous works and others are modified. We do not claim the operations are our contribution. Instead, our contribution lies in figuring out which operations we need and how to use them to build an effective network. Section 4.2 consists of the proposed attention and its equivariant architecture and Section 4.3 shows what Transformers look like after adopting the chosen equivariant operations. Since our contribution includes choosing the right equivariant operations, what is affected by this choice should be included in the same section.
>
> Additionally, the writing follows the structure of previous works on equivariant Transformers [1, 2] as well as previous works on Transformers in computer vision [3] and graph [4].
>
> Reference:
>
> [1] Fuchs et al. SE(3)-Transformers: 3D Roto-Translation Equivariant Attention Networks. NeurIPS 2020.
>
> [2] Thölke et al. Equivariant Transformers for Neural Network based Molecular Potentials. ICLR 2022 Spotlight.
>
> [3] Dosovitskiy et al. An Image is Worth 16x16 Words: Transformers for Image Recognition at Scale. ICLR 2021 Oral.
>
> [4] Ying et al. Do Transformers Really Perform Badly for Graph Representation? NeurIPS 2021.
>
> ---
> > 3. In Sec. 4, the analyses of equivariance are missing. Why do the proposed components still keep equivariance?
>
> Equivariant operations can propagate the transformation (we consider E(3) transformation in this work) in input vector spaces to output spaces by design and satisfy the definition of equivariance: $f(D_X(g) X) = D_Y(g) f(X)$ where f is an equivariant function mapping between input vector space $X$ and output vector space $Y$ and $D_X$ and $D_Y$ are transformation matrices in $X$ and $Y$ parametrized by $g$. Below we analyze how each operation in Section 4.1 remains equivariant.
>
> - Linear.
>
>     Since one output type-L vector is a linear combination of other input type-L vectors, which are transformed by the same matrix $D_X(g)$, the output type-L vector is transformed by the same matrix, meaning that $D_X(g) = D_Y(g)$.
>
>
> - Layer Normalization.
>
>     For scalar parts (L = 0), they are always the same regardless of E(3) transformations, and thus we can apply any function. For non-scalar parts (L > 0), the L2 norm of any type-L vector is invariant to E(3) transformations. Therefore, the scaling of dividing by the root mean square value (RMS) of L2 norm and multiplying by a learnable parameter $\gamma$ remains the same under E(3) transformations. Multiplying an equivariant feature with an invariant number results in an equivariant feature, and therefore the operation is equivariant.
>
> - Gate.
>
>     Similar to Layer Normalization, we can apply any function to the scalar part (L = 0). We apply SiLU to the first $C_0$ channels of the scalar part and sigmoid to other channels to obtain non-linear weights. For the non-scalar part (L > 0), we multiply each type-L vector with its corresponding non-linear weight. Since the non-linear weights are invariant, multiplying equivariant features with those non-linear weights results in equivariant features.
>
> - Depth-wise Tensor Product.
>
>     This operation is based on equivariant tensor product operations and restricts that one channel in output irreps feature depends on one channel in input irreps features. The one-to-one dependence of channels does not change the interaction of different type-L vectors, and therefore, the operation is equivariant.
>
> Since each module in Figure 1 is built from equivariant operations described in Section 4.1, the transformation acting on the input of each module can propagate to the output, making each module equivariant. Similarly, since the whole network consists of equivariant modules and operations, the network is equivariant. We will add a more detailed description to the appendix.

---

> > ### Author Response · Authors · 2022-11-15
> > **Response to Reviewer 1yMo (2/2)**
> >
> > > 4. It seems that the performances on MD17 and QM9 are marginal.
> >
> > Below are some potential reasons that the performance gain on MD17 and QM9 is not significant:
> >
> > 1. MD17 and QM9 are well-explored datasets and could be overfitted by many recent methods as mentioned by Reviewer **d6yC**.
> >
> > 2. The datasets are small and contain less diverse atom types. Thus, it would not necessarily require more expressive models. We have already observed this in our ablation studies in Section 5.4.
> >
> > 3. It is unclear to know how accurate we can be for each dataset and what is the limit of accuracy. If the model is achieving the lowest errors we can have based on the training set, the performance should be considered significant rather than marginal.
> >
> >
> > **We would like to emphasize that Equiformer simultaneously achieves the best results on QM9 and MD17.**
> > In contrast, although TorchMD-NET [1] and PaiNN [2] achieve competitive results on QM9 (Table 1), they are clearly outperformed by NequIP [3] and Equiformer on MD17 (Table 2). Besides, NequIP does not report results on QM9, and based on our ablation study (Table 7), more expressive architectures can achieve better results, indicating that NequIP with linear messages would be outperformed by Equiformer with MLP attention and non-linear messages on OC20 dataset.
> >
> > Reference:
> >
> > [1] Thölke et al. Equivariant Transformers for Neural Network based Molecular Potentials. ICLR 2022 Spotlight.
> >
> > [2] Schutt et al. Equivariant message passing for the prediction of tensorial properties and molecular spectra. ICML 2021.
> >
> > [3] Batzner et al. E(3)-equivariant graph neural networks for data-efficient and accurate interatomic potentials. Nature Communications 2022.
> >
> > ---
> > > 5. Why are the baselines evaluated on MD17, QM9, and OC20 different?
> >
> > It is simply because not all models are extensively evaluated on the three different datasets. However, we have compared Equiformer to highly competitive models on each dataset and shown that Equiformer simultaneously achieves the best results on all the datasets.

---

> > > ### Comment · Reviewer_1yMo · 2022-11-15
> > > **Please revise the manuscript according to your feedback**
> > >
> > > Some feedback sounds reasonable. But I still encourage the authors to revise their manuscript according to the feedback and highlight the changes with different color before I make the judgement for the rebuttal.
> > >
> > > Thanks,

---

> > > > ### Author Response · Authors · 2022-11-16
> > > > **Update Manuscript**
> > > >
> > > > Thank you for your quick response!
> > > >
> > > > We have updated the manuscript based on comments from all reviewers.
> > > >
> > > > Please let us know if you have any further question.

---

> > > ### Comment · Reviewer_1yMo · 2022-11-17
> > > **Add average MAE for energy and forces on MD17?**
> > >
> > > Maybe you could add average MAE for energy and forces on MD17 through all the molecules.

---

> > > > ### Author Response · Authors · 2022-11-24
> > > > **Response to "Add average MAE for energy and forces on MD17"**
> > > >
> > > > We are not sure we understand what the reviewer is asking. We cannot average the MAE of forces + energy as they are not in the same units and we also cannot average the MAE of forces for all molecules because 1) that's not a standard way of reporting results for this benchmark and 2) different molecules have different ranges of errors so an average MAE across all molecules is not particular meaningful.

---

> > ### Comment · Reviewer_1yMo · 2022-11-17
> > **The analysis of equivariance for Linear, Layer Normalizatoin, Gate, and Depth-wise Tensor Product**
> >
> > My major concern is the writing of the paper but not the contribution of the paper. The previous version is hard to follow because all the parts are mixed up. The revised version shortly summarizes each part of Sec 4 and make it more clear.
> >
> > On the other hand, some navigation to C.1 should be stated  in Sec 4.1.

---

> > > ### Author Response · Authors · 2022-11-18
> > > **Updating Manuscript to Include Navigation to Sec. C.1 in Sec. 4.1**
> > >
> > > We thank the review for acknowledging the revised version is more clear and increasing the score from 5 to 6. We include the navigation to Sec. C.1 in Sec. 4.1 and update the manuscript. Thanks for pointing out this.

---

> ### Author Response · Authors · 2022-12-07
> **Please let us know if you have other questions or comments**
>
> We believe we have addressed your concerns, especially the two comments (**the analysis of equivariance** and **average MAE for energy /force on MD17 dataset**) posted on November 17th.
>
> Please let us know if you have other questions or comments and we will be very happy to respond.

---

### Official Review · Reviewer_d6yC · 2022-10-31

**Confidence:** 4
**Correctness:** 3
**Technical Novelty And Significance:** 2
**Empirical Novelty And Significance:** 3
**Recommendation:** 6

**Clarity, Quality, Novelty And Reproducibility:**

As stated above, this paper did a good job on explaining and comparing the difference between different equivariant models and transformers in Related Work. However, there are still certain concerns for the presentation.

1. Section 4 only focuses on SE(3) equivaraince. What about E(3) equivariance? By the way, SH is SE(3)-equivariant not E(3).

2.  It is hard to justify the sufficient novelty by the current presentation.


**Strength And Weaknesses:**

Strengths:

1. This paper did a good job on explaining and comparing the difference between different equivariant models and transformers: TFN, SE3-Transformer, NeuIP, SEGNN.

2. Besides QM9 and MD17 (which are actually well-explored datasets and could be overfitted by many recent methods), the authors evaluate their method on OC20 (the direct setting), a new but more desirable dataset for performance comparison. It is valuable to see that the proposed method achieves good results on IS2RE. Necessary ablation studies are also performed.

Weaknesses:

1. I totally understand that MLP attention + non-linear message is the most desirable combination as supported by this paper. Extending current methods to the form proposed in this paper is not that surprising. The authors are suggested to futher highlight the novelty of the proposed method. The current writing makes people think that it is just a simple combination of serveral components (Sec. 4). Readers are interested in knowing which part is novel and which part is previously proposed.

2.  The ablation studies are insufficient. From Table 6-7, it seems the contribution of this paper only lies in the form of the attention it used. If this is the case, the contribution is weak. In other words,  the authors are suggested to perform more experiments to show what other novelty (other than the attention) is and why it works.

3. For the OC experiments, did you try the relaxation setting?

**Summary Of The Paper:**

This paper proposes an E(3)/SE(3) equivariant transformer on 3D molecular graphs. The central point is to apply MLP attention + non-linear message for better capturing the interaction between atoms.  The attention weights are computed by a non-linear MLP attention mechanism based on the type-0 irreps features. Then the attention weights are multiplicated with other irreps features with type >0. The evaluations are carried out on QM9, MD17 and OC20, which supports the benefit of the proposed method.

**Summary Of The Review:**

Overall, this paper has made some variable efforts on improving current equivariant GNN models, by considering the techiniques from Transformers.  However, for its current presentation, it seems the technical novelty is marginal and the experimental evaluations are still insufficient to support the contributions.

---

> ### Author Response · Authors · 2022-11-15
> **Response to Reviewer d6yC (1/2)**
>
> We thanks the reviewer for helpful feedback and address the comments below.
>
> ---
> > 1. Minor clarification of the sentence “Then the attention weights are multiplicated with other irreps features with type >0” in the summary of the paper.
>
> The attention weights are multiplied with either $f_{ij}^{(L)}$ (linear message) or $v_{ij}$ (non-linear message) and both $f_{ij}^{(L)}$ and $v_{ij}$ contain not only type-L vectors with L > 0 but also scalars (L = 0).
>
> ---
> > 2. [Weakness 1] Extending current methods to the form (MLP attention + non-linear messages) proposed in this paper is not that surprising.
>
> We would like to emphasize that the proposed MLP attention + non-linear messages can improve upon linear messages (used by NequIP) even on small datasets like MD17, which contains only 950 conformations of the same type of a molecule in the training set. This empirical result would be non-trivial since some works on deeper and more expressive GNNs report that more expressive networks would not always improve upon simple GNNs on small datasets [1].
>
> Reference:
>
> [1] Addanki et al. Large-scale graph representation learning with very deep GNNs and self-supervision. 2022.
>
> ---
> > 3. [Weakness 1] The authors are suggested to further highlight the novelty of the proposed method.
>
> Please see **General Response 1** for the contribution of Equiformer compared to other equivariant Transformers. We summarize the novelties of this work in **General Response 2** and highlight the differences from other equivariant Transformers in **General Response 3**.
>
> ---
> > 4. [Weakness 1] The current writing makes people think that it is just a simple combination of several components (Sec. 4). Readers are interested in knowing which part is novel and which part is previously proposed.
>
> We do not consider a simple combination to be a weakness. That the combination appears simple is an intentional result. We have figured out we only need four well-defined operations mentioned in Section 4.1 to build an effective architecture, and each part of our proposed equivariant graph attention has motivations as described in Section 4.2. Moreover, it is simple and effective: Equiformer achieves the best results for all datasets considered and circumvents all the problems in other equivariant Transformers (please see **General Response 3**).
>
> Please see **General Response 4** for detailed description of which part is novel and which part is previously proposed.
>
> ---
>
> > 5. [Weakness 2] From Table 6-7, it seems the contribution of this paper only lies in the form of the attention it used. The authors are suggested to perform more experiments to show what other novelty (other than the attention) is and why it works.
>
> We summarize the novelties of this work in **General Response 2**. Another novelty lies in how we choose the right operations as well as internal representations (vectors of any degree L) and combine them in an effective manner to build a simple and effective architecture, said Equiformer with dot product attention and linear messages. The proposed attention further improves the performance of this architecture.
>
> As for experiments, since the architecture, Equiformer with dot product attention and linear messages, is proposed by only replacing original operations with equivariant counterparts and including tensor products, we make few modifications to the original Transformer architecture and thus do not need experiments to justify our decisions. The only difference from previous equivariant Transformers probably lies in that we use equivariant layer normalization mentioned in Section 4.1, which is empirically better than the norm nonlinearities used by SE(3)-Transformer [1]. The comparison can be found in **General Response 3** and justify our decision. Besides, since the proposed attention combining MLP attention and non-linear messages is what differs significantly from original Transformers, we conduct extensive ablation studies to verify the design decision.
>
> Reference:
>
> [1] Fuchs et al. SE(3)-Transformers: 3D Roto-Translation Equivariant Attention Networks. NeurIPS 2020.

---

> > ### Author Response · Authors · 2022-11-15
> > **Response to Reviewer d6yC (2/2)**
> >
> > > 6. For the OC experiments, did you try the relaxation setting?
> >
> > No, we did not try the relaxation setting due to the huge computational cost and note that most previous or concurrent works do not use the relaxation setting [1, 2, 3].
> >
> > The relaxation setting requires training for the task of S2EF (structure to energy and forces), and the smallest subset of S2EF dataset used for relaxation contains 2M graphs, which is about $5\times$ larger than the size of IS2RE training set. The default number of training epochs is $5\times$  larger than that of IS2RE. These result in at least  $20$ $( = 5 \times 4 )$ times more computation and can take around 80 GPU-days, which is not affordable to most people.
> >
> > Reference:
> >
> > [1] Liu et al. Spherical Message Passing for 3D Molecular Graphs. ICLR 2022.
> >
> > [2] Brandstetter et al. Geometric and Physical Quantities improve E(3) Equivariant Message Passing. ICLR 2022 Spotlight.
> >
> > [3] Wang et al. ComENet: Towards Complete and Efficient Message Passing for 3D Molecular Graphs. NeurIPS 2022.
> >
> > ---
> > > 7. Section 4 only focuses on SE(3) equivariance. What about E(3) equivariance?
> >
> > For the case of E(3) equivariance, we can simply use the same architecture as described in Figure 1 and change internal representations from type-L vectors to type-(L, p) vectors as mentioned in Section A.3. In Section A.3, we mention that when E(3) equivariance is considered, the operations Linear and Layer Normalization will treat type-(L, e) vectors and type-(L, o) vectors as different types. This means that we linearly combine or normalize these two types of vectors in a separate manner. In Section A.4, we discuss how tensor products behave when E(3) equivariance is considered. For Gate, we extend our discussion in Section 4.1 below.
> >
> > Given input $x$ containing non-scalar $C_{(L, p)}$ type-$(L, p)$ vectors with $0 < L \leq L_{max}$ and $p \in ${$e, o$}, $C_{(0, o)}$ pseudo-scalars and $(C_{(0, e)} + C_{(0, o)} + C_{(L, p)})$ scalars, we apply SiLU to the first $C_{(0, e)}$ type-$(0, e)$ vectors (scalars) and sigmoid function to the other $(C_{(0, o)} + C_{(L, p)} )$ type-$(0, e)$ vectors to obtain non-linear weights and
> > multiply each pseudo-scalar or type-$(L, p)$ vector with corresponding non-linear weights. After the gate activation, the number of channels for type-$(0, e)$ vectors is reduced to $C_{(0, e)}$. In short, when E(3) equivariance is considered, the original scalar parts correspond to type-$(0, e)$ vectors, but type-$(0, o)$ (pseudo-scalars) are treated in the same manner as other vectors of higher degree.
> >
> > Thanks for pointing out this and we will add a more detailed description to the appendix.
> >
> > ---
> > > 8. By the way, SH is SE(3)-equivariant not E(3).
> >
> > Spherical harmonics (SH) are E(3)-equivariant. As discussed in Section A.3, SH of degree L transforms in the same manner as type-(L, p) vectors, where p is odd if L is odd and p is even if L is even. Since the behavior of SH is well-defined under inversion, SH is equivariant to inversion and thus E(3)-equivariant.

---

> > > ### Comment · Reviewer_d6yC · 2022-11-17
> > > **More explanations are needed**
> > >
> > > Thank the authors for the efforts. The feedbacks help address some concerns. Yet, there are still some concerns.
> > >
> > > 1. Regarding General Response 4, please add the experimental comparison to justify the difference between the proposed Depth-wise Tensor Products  and fully connected tensor products used in SE(3)-Transformer.
> > >
> > > 2. Would you please comment more on why MLP attention is better than Dot-product attention on the datasets in this paper?
> > >
> > > 3. Spherical harmonics are SE(3)-equivariant not E(3)-equivariant. Inversion is not equal to Reflection, so SH is not E(3)-equivariant even it is equivariant to inversion. Please correct the related presentations.

---

> > > > ### Author Response · Authors · 2022-11-20
> > > > **Response to Reviewer d6yC (Part 3)**
> > > >
> > > > We address the additional comments below and start with number 9 for consistency with previous responses (i.e., we use 9 to refer to the question 1: Regarding General Response 4, please add the experimental comparison...).
> > > >
> > > > ---
> > > > > 9. Regarding General Response 4, please add the experimental comparison to justify the difference between the proposed Depth-wise Tensor Products and fully connected tensor products used in SE(3)-Transformer.
> > > >
> > > > We will conduct experiments on QM9 and provide the results within a week.
> > > >
> > > > **[Update on 11/23]**
> > > >
> > > > The comparison between depth-wise tensor products and fully connected tensor products is shown below. Using depth-wise tensor products results in lower errors and less training time than using fully connected tensor products.
> > > >
> > > > |                                             | $\alpha$ | $\Delta \epsilon$ | $\epsilon_{HOMO}$ | $\epsilon_{LUMO}$ |  $\mu$ | $C_{\nu}$ | Training time (mins/epoch) |
> > > > |---------------------------------------------|:------:|:---------------:|:----------------:|:----------------:|:----:|:-------:|----------------------------|
> > > > | Equiformer (fully connected tensor product) |  .061  |        35       |        20        |        18        | .016 |   .034  |            14.6            |
> > > > | Equiformer  (depth-wise tensor product)     |  .046  |       30        |        15        |        14        | .011 |   .023  |            12.2            |
> > > >
> > > > ---
> > > >
> > > > > 10. Would you please comment more on why MLP attention is better than dot product attention on the datasets in this paper?
> > > >
> > > > The reason that MLP attention can be theoretically stronger than dot product attention is that MLP attention uses non-linear functions to generate attention weights while dot product attention uses linear functions (i.e., dot product). Non-linear functions are theoretically more expressive than linear functions, and therefore MLP attention can be better than dot product attention. The argument is the same as why non-linear messages are better than linear messages.
> > > >
> > > > Please let us know what part is not clear if possible.
> > > >
> > > > ---
> > > >
> > > > > 11. Spherical harmonics are $SE(3)$-equivariant not $E(3)$-equivariant. Inversion is not equal to Reflection, so SH is not $E(3)$-equivariant even if it is equivariant to inversion. Please correct the related presentations.
> > > >
> > > > Could you please give an example for why you believe the spherical harmonics are not equivariant under $E(3)$ but are for $SE(3)$?
> > > >
> > > > If your concern is specifically about mirror (reflection) symmetries, we need only talk about the group $O(3)$ which is the group of inversion and rotation, and therefore the composition of rotations and inversion - roto-inversions e.g. mirror (reflection). As $O(3)$ is a direct product of $SO(3)$ and $Z_2$, irreducible representations (irreps) of $O(3)$ carry two indices: $L$ (same $L$ as for $SO(3)$) and an index indicating the symmetry of the irrep under inversion (even or odd), also known as the parity of the irrep. Spherical harmonics transform as specific $O(3)$ irreps: spherical harmonics that have even $L$ are even under inversion and those that have odd $L$ are odd under inversion. By incorporating the multiplication rules for these irreps as we do in our tensor product operations which includes treating irreps with the same $L$ but different parity as distinct types, we achieve equivariance. Note that the only difference between $SO(3)$ and $O(3)$ tensor products is that we have to add the following multiplication rules for even (e) and odd (o) $Z_2$ irreps: $e \times e \rightarrow e, o \times e \rightarrow o, e \times o \rightarrow o, o \times o \rightarrow e$.
> > > >
> > > >
> > > > In fact, spherical harmonics can be generated by taking tensor products of Euclidean vectors (transformed by $L = 1, p = o$ irreps). Since the tensor products and Euclidean vectors are $E(3)$-equivariant, so are the spherical harmonics. Additionally, this is how the library e3nn implements spherical harmonics.
> > > >
> > > >
> > > > Please let us know if this comment has not addressed your concern regarding the equivariance of spherical harmonics.

---

> ### Author Response · Authors · 2022-12-07
> **Please let us know if you have other questions or comments**
>
> We believe we have addressed your concerns, particularly:
>
>   1. [**9. in Response to Reviewer d6yC (Part 3)**]  the comparison between depth-wise tensor products and fully connected tensor products and
>
>   2. [**11. in Response to Reviewer d6yC (Part 3)**] that spherical harmonics are $E(3)$-equivariant .
>
> Please let us know if you have other questions or comments. We will be very happy to respond.

---

### Official Review · Reviewer_sfAh · 2022-11-02

**Confidence:** 4
**Correctness:** 4
**Technical Novelty And Significance:** 2
**Empirical Novelty And Significance:** 3
**Recommendation:** 8

**Clarity, Quality, Novelty And Reproducibility:**

The paper is well written and the related literature is sufficiently discussed.
The complex architectures proposed are much easier to understand thanks to the clear figures.

The supplementary materials describe in details the experiments performed and the architectures.


**Strength And Weaknesses:**



While the main theoretical contribution is limited to combining existing methods, I think the paper has high practical relevance.
Indeed, besides the state-of-the-art results, the extensive experimental section provides insights about benefits and costs of different sub-modules.

Even if many related works often don't do it, I encourage the authors to report mean and std of their results, aggregated over multiple runs (the difference in performance in some experiments is relatively small).
I think this is especially important in the ablation study: a more statistically sound comparison would increase the impact of this work.


**Summary Of The Paper:**

The paper proposes a new equivariant neural network to process molecular graphs.
The authors successfully combine two already effective equivariant architecture designs - i.e. non-linear message passing and transformers - and achieve improved performance and computational gains on multiple datasets and tasks.
Moreover, a large ablation study is included to validate the importance of each contribution.


**Summary Of The Review:**

SUMMARY

The work combines existing methods in a new, successful way and is supported by strong empirical results.
The authors could improve the work by proving the statistical significance of some results (e.g. reporting mean and std over multiple runs of their experiments).



OTHER QUESTION


You argue that (1) the MLP attention is more computationally efficient and more expressive than (2) Dot-Product attention.
In C.4, you mention that the higher cost in (2) is partially because the query and the key need to have the same dimension of the value.
Why can't smaller keys and queries be used?
If only frequency-0 keys and queries were used in (2), as is done in (1), would its computational cost be lower than (1) ?


The description of radial basis and radial functions at page 6 is a bit unclear. What does it mean that the function consists of *two* MLPs? Do you mean a 2-layers MLP?
Also, I am not sure I understood the first part of this sentence:
"[We transform radial basis with a learnable radial function] to generate weights for those DTP layers."

Since this work builds mostly on top of SEGNN and SE(3)-Transformer, isn't it better to use the train/val/test split used by those works in the QM9 experiments?
Do you think the different split could make a significative difference in the results?

---

> ### Author Response · Authors · 2022-11-15
> **Response to Reviewer sfAh (1/2)**
>
> We thank the reviewer for helpful feedback and address the comments below.
>
> ---
> > 1. Mean and standard deviation of results in ablation studies.
>
> We train Equiformer with MLP attention and non-linear message passing with 3 different random seeds on QM9 and OC20. We summarize the results below and compare with other models in ablation studies. For both QM9 and OC20, the statistical variances are smaller than the performance gain.
>
> - QM9 (mean absolute errors (MAE)):
> | Index | Non-linear message | MLP attention | Dot product attention | $\alpha$              | $\Delta \epsilon$ | $\epsilon_{HOMO}$ | $\epsilon_{LUMO}$ | $\mu$                 | $C_{\nu}$                 |
> |-------|--------------------|---------------|-----------------------|---------------------|-----------------|-----------------|-----------------|---------------------|---------------------|
> | 1     | Y                  | Y             |                       | 0.045726 ± 0.000269 | 29.69  ± 0.29   | 15.39  ± 0.28   | 14.21  ± 0.15   | 0.011333 ± 0.000241 | 0.023520 ± 0.000399 |
> | 2     |                    | Y             |                       | .051                | 32              | 16              | 16              | .013                | .025                |
> | 3     |                    |               | Y                     | .053                | 32              | 17              | 16              | .013                | .025                |
>
> - OC20
>     - Energy MAE
> | Index | Non-linear message | MLP attention | Dot product attention | ID          | OOD Ads     | OOD Cat     | OOD Both    | Avg.        |
> |-------|--------------------|---------------|-----------------------|-------------|-------------|-------------|-------------|-------------|
> | 1     | Y                  | Y             |                       | 0.5063 ± 0.0018 | 0.6251 ± 0.0014 | 0.5029 ± 0.0018 | 0.5584 ± 0.0056 | 0.5482 ± 0.0012 |
> | 2     |                    | Y             |                       | 0.5168          | 0.6308          | 0.5088          | 0.5657          | 0.5555          |
> | 3     |                    |               | Y                     | 0.5386          | 0.6382          | 0.5297          | 0.5692          | 0.5689          |
>     - EwT
> | Index | Non-linear message | MLP attention | Dot product attention | ID          | OOD Ads     | OOD Cat     | OOD Both    | Avg.        |
> |-------|--------------------|---------------|-----------------------|-------------|-------------|-------------|-------------|-------------|
> | 1     | Y                  | Y             |                       | 5.07 ± 0.14 | 3.00 ± 0.09 | 5.00 ± 0.08 | 3.09 ± 0.09 | 4.04 ± 0.09 |
> | 2     |                    | Y             |                       | 4.59        | 2.82        | 4.79        | 3.02        | 3.81        |
> | 3     |                    |               | Y                     | 4.37        | 2.60        | 4.36        | 2.86        | 3.55        |
>
> ---
>
> > 2. The comparison of efficiency between (1) MLP attention and (2) dot-product attention. In C.4, the higher cost in (2) is partially because query and key need to have the same dimension as value. Why can't smaller keys and queries be used? If only frequency-0 (scalar part) keys and queries were used in (2), as is done in (1), would its computational cost be lower than (1) ?
>
> We can use smaller keys and queries as in MLP attention. However, this modification does not follow the practice of many Transformers, which always keep queries, keys and values of the same dimension. We report the computational time on QM9 and OC20 when only the scalar part (type-0 or frequency-0) of queries and keys is used in dot-product attention below. We use 1 V100 GPU for QM9 and 2 V100 GPUs for OC20. The training time of dot product attention is still slower, and the reasons are as follows. It requires one extra linear operation to generate queries vectors and additional duplicating queries vectors for taking dot products with key vectors. We note that if only the scalar part is used for dot product attention, the features of higher degree L in source nodes (generating queries) will be ignored, which can lead to worse performance.
>
> |                                                    | QM9 training time (mins/epoch) | OC20 training time (mins/epoch) |
> |----------------------------------------------------|:------------------------------:|:-------------------------------:|
> | MLP attention                                      |              7.11              |                97               |
> | Dot product attention with scalar queries and keys |              7.28              |               100               |

---

> > ### Author Response · Authors · 2022-11-15
> > **Response to Reviewer sfAh (2/2)**
> >
> > > 3. Clarification of radial functions. What does it mean that the function consists of two MLPs? Do you mean a 2-layers MLP? Also, I am not sure I understood the first part of this sentence: "[We transform radial basis with a learnable radial function] to generate weights for those DTP layers."
> >
> > Yes, we mean a 2-layer MLP. More specifically, the radial function consists of two linear layers, each followed by layer normalization (LN) and SiLU, and then a final linear layer. We will update our manuscript. Please see `nets/radial_func.py` in the anonymous repo for implementation details.
> > As for “transform radial basis with a learnable radial function”, it means we feed the radial basis to the radial function as input and take the output of the radial function as the weights for DTP layers. Please see Line 241 - Line 244 in `nets/graph_attention_transformer.py` in the anonymous repo for implementation details.
> >
> > ---
> >
> > > 4. Comparison to SEGNN and SE(3)-Transformer on QM9 with the same training, validation and testing split.
> >
> > We train Equiformer on the first six tasks of QM9 and use the same training, validation and testing split as SE(3)-Transformer and SEGNN. We note that the split used by SE(3)-Transformer and SEGNN has 100k molecules in the training set, and the split we use in our paper has 110k molecules. Thus, using the same split as SE(3)-Transformer and SEGNN leads to slightly worse results (higher errors). The comparison of mean absolute errors (MAE) is shown below. Equiformer outperforms SE(3)-Transformer and SEGNN by clear margins when the same split is used.
> >
> > |                                                        | $\alpha$ | $\Delta \epsilon$ | $\epsilon_{HOMO}$ | $\epsilon_{LUMO}$ |  $\mu$ | $C_{\nu}$ |
> > |--------------------------------------------------------|:------:|:---------------:|:----------------:|:----------------:|:----:|:-------:|
> > | SE(3)-Transformer                                      | .142   | 53              | 35               | 33               | .051 | .054    |
> > | SEGNN                                                  | .060   | 42              | 24               | 21               | .023 | .031    |
> > | Equiformer (same split as SE(3)-Transformer and SEGNN) | .056   | 33              | 17               | 16               | .014 | .025    |
> > | Equiformer  (split in Table 1)                         | .046   | 30              | 15               | 14               | .011 | .023    |

---

> ### Author Response · Authors · 2022-12-07
> **Please let us know if you have other questions or comments**
>
> We believe we have addressed your concerns.
> Please let us know if you have other questions or comments.
> We will be very happy to respond.

---

### Author Response · Authors · 2022-11-15
**General Response 1: Contribution of Equiformer compared to other equivariant Transformers**

We compare the results and impact of Equiformer with those of other equivariant Transformers in the following four aspects:
1. Previous equivariant Transformers do not perform well across datasets. For instance, SE(3)-Transformer [1] is not as performant as other equivariant networks on QM9 (Table 1). TorchMD-NET [2] does not achieve comparable results to NequIP [3] on MD17 (Table 2) although it is competitive on QM9 (Table 1).
2. Equiformer simultaneously achieves the best results for MD17, QM9 and OC20 datasets, indicating that the Transformer architecture is generally effective in the literature of equivariant neural networks and 3D atomistic graphs.
3. Extensive ablation studies have been conducted to justify a better attention mechanism in this domain.
4. To the best of our knowledge, we are the first to apply equivariant Transformers to large and complicated datasets like OC20 and demonstrate that equivariant Transformers can achieve competitive results to large models like GNS [4] and Graphormer [5] while saving 2.3X to 15.5X training time (Table 5).

We will add this to a new subsection in related works in appendix.


Reference:

[1] Fuchs et al. SE(3)-Transformers: 3D Roto-Translation Equivariant Attention Networks. NeurIPS 2020.

[2] Thölke et al. Equivariant Transformers for Neural Network based Molecular Potentials. ICLR 2022 Spotlight.

[3] Batzner et al. E(3)-equivariant graph neural networks for data-efficient and accurate interatomic potentials. Nature Communications 2022.

[4] Godwin et al. Simple GNN Regularisation for 3D Molecular Property Prediction & Beyond. ICLR 2022.

[5] Ying et al. Do Transformers Really Perform Badly for Graph Representation? NeurIPS 2021.

---

### Author Response · Authors · 2022-11-15
**General Response 2: Novelty of this work**

The novelty of this work lies in the two aspects:

**(a)** A new equivariant Transformer, Equiformer with dot product attention and linear messages (index 3 in Table 6 and 7), is proposed by only replacing original operations in Transformers with equivariant counterparts and including tensor products. Although with minimal modifications to the Transformer architecture, it surprisingly achieves better performance than other equivariant networks. Moreover, it is simpler, more general and more efficient than other equivariant Transformers. We extend our discussion in related works and present the detailed difference from other equivariant Transformers in **General Response 3**.

**(b)** A novel attention mechanism combining MLP attention and non-linear messages and the corresponding equivariant version, which further improves the performance of the architecture in (a). In contrast, previous works combine message passing and attention by merely stacking message-passing layers and Transformer blocks [1, 2]. This requires tuning the numbers of message-passing layers and Transformer blocks and might inherit the issues of message passing such as oversmoothing [3].

The final proposed Equiformer model combines **(a)** and **(b)**. We will make **(a)** more clear in the paper.

Reference:

[1] Wu et al. Representing Long-Range Context for Graph Neural Networks with Global Attention. NeurIPS 2021.

[2] Rampášek et al. Recipe for a General, Powerful, Scalable Graph Transformer. NeurIPS 2022.

[3] Kim et al. Pure Transformers are Powerful Graph Learners. NeurIPS 2022.

---

### Author Response · Authors · 2022-11-15
**General Response 3: Detailed differences of the proposed architecture in (a) in General Response 2 from other equivariant Transformers.**

We extend our discussion in related works and discuss the differences between the proposed architecture mentioned in **(a)** in **General Response 2** and other equivariant Transformers [1, 2, 3]. We note that **the proposed architecture has not been explored, and it is (1) simpler and (2) more general and contains (3) more efficient tensor product operations.**

(1) Simpler: We simply replace original operations with their equivariant counterparts and include tensor products without making further modifications. In contrast, [1] merges normalization and activation to form norm nonlinearities, which does not exist in original Transformers. We empirically show that the norm nonlinearities [1] leads to higher errors compared to the equivariant layer norm used by our work and summarize below the results when either dot product attention and linear messages or MLP attention and non-linear messages are used.

(2) More general: [1] and our proposed architecture can support vectors of any degree L while [2, 3] are limited to L = 0 and 1. It has been shown that higher L (e.g., L up to 2 and 3) can improve the performance of networks on QM9 [4] and MD17 [5]. Thus, the incapability to use L higher than 1 can limit the performance of those equivariant Transformers [2, 3].

(3) More efficient tensor products: compared to [1], we use more efficient depth-wise tensor products instead of fully connected tensor products. Since the proposed architecture and [1] use relative distances to parametrize the weights of tensor products, depth-wise tensor products allow more channels without incurring out-of-memory errors. Specifically, for QM9, [1] only uses 16 channels for vectors of each degree L while the proposed architecture can use 128, 64, and 32 channels for vectors of degree 0, 1, and 2. Using a very small number of channels can potentially lead to insufficient model capacity.

We summarize that **the novelty of the proposed architecture lies in how we choose the right operations as well as internal representations (vectors of any degree L) and combine them in an effective manner that achieves the three advantages mentioned above.** We will add this to a new subsection in related works in appendix.

---

|                                                   | $\alpha$ | $\Delta \epsilon$ | $\epsilon_{HOMO}$ | $\epsilon_{LUMO}$ |  $\mu$ | $C_{\nu}$ |
|---------------------------------------------------|:------:|:---------------:|:----------------:|:----------------:|:----:|:-------:|
| Norm nonlinearities [1]                           | .053   | 34              | 18               | 17               | .016 | .027    |
| Equivariant layer normalization used in this work | .053   | 32              | 17               | 16               | .013 | .025    |

Table: Comparison of mean absolute error (MAE) between norm nonlinearities and equivariant layer normalization when dot product attention + linear message is used.

---

|                                                   | $\alpha$ | $\Delta \epsilon$ | $\epsilon_{HOMO}$ | $\epsilon_{LUMO}$ |  $\mu$ | $C_{\nu}$ |
|---------------------------------------------------|:------:|:---------------:|:----------------:|:----------------:|:----:|:-------:|
| Norm nonlinearities [1]                           | .046   | 32              | 16               | 15               | .014 | .024    |
| Equivariant layer normalization used in this work | .046   | 30              | 15               | 14               | .011 | .023    |

Table: Comparison of mean absolute error (MAE) between norm nonlinearities and equivariant layer normalization when MLP attention + non-linear message is used.

---

Reference:

[1] Fuchs et al. SE(3)-Transformers: 3D Roto-Translation Equivariant Attention Networks. NeurIPS 2020.

[2] Thölke et al. Equivariant Transformers for Neural Network based Molecular Potentials. ICLR 2022 Spotlight.

[3] Le et al. Equivariant graph attention networks for molecular property prediction. 2022.

[4] Brandstetter et al. Geometric and Physical Quantities improve E(3) Equivariant Message Passing. ICLR 2022 Spotlight.

[5] Batzner et al. E(3)-equivariant graph neural networks for data-efficient and accurate interatomic potentials. Nature Communications 2022.

---

### Author Response · Authors · 2022-11-15
**General Response 4: Detailed description of which components in Section 4 are novel and which are previously proposed**

1. Equivariant Operations for Irreps Features (Section 4.1): The operations used are either previously proposed or modified from similar operations in previous works. We note that **our contribution is to figure out the right operations to build a simple and effective network as claimed instead of developing new operations.**

    a. Linear and Gate operations directly come from previous works as mentioned in our work.

    b. Layer Normalization is directly extending original Layer Normalization for scalars to support vectors of different types.

    c. Depth-wise Tensor Products can be viewed as an extension of fully connected tensor products used in SE(3)-Transformer. We modify the dependence of output channels and restrict one output channel to depend on one input channel. The reason to do that is simply to make the computation of tensor products more affordable. We mention “depth-wise convolution” is to simply draw an analogy to make this concept more understandable.

2. Equivariant Graph Attention (Section 4.2): **The idea of combining MLP attention and non-linear messages is novel.** It is based on our observation that the messages can be decomposed into attention weights and value vectors as in Equation (2). This observation motivates using more expressive functions for both attention weights and value vectors to achieve overall better expressiveness. Moreover, **an effective equivariant version of the combination of MLP attention and non-linear messages is proposed** and based on the operations in Section 4.1.

3. Overall Architecture (Section 4.3): This part is included for completeness and is based on what has been proposed in previous works. This shows what Transformers look like after including equivariant operations mentioned in Section 4.1.

---

### Author Response · Authors · 2022-11-16
**Update Manuscript**

We incorporate the helpful comments from reviewers and update the manuscript.

We highlight the changes in blue in our manuscript and summarize them below:

1. (**General Response 1**) We add the contribution of Equiformer compared to other equivariant Transformers in Sec. B.2.

2. (**General Response 2**) We highlight the novelty of this work in the sections of abstract, introduction, method and experiment. We mention that the novelties lies in **(a)** Equiformer with dot product attention and linear message and **(b)** equivariant graph attention.

3. (**General Response 3**) We add the detailed differences of the proposed architecture in **(a)** in **General Response 2** from other equivariant Transformers in Sec. B.2.

4. (**Response to Reviewer d6yC - 7**) We add a detailed description of incorporating E(3)-equivariance in Sec. C.4.

5. (**Response to Reviewer 1yMo - 3**) We add an analysis on how each equivariant operation in Sec. 4.1 remains equivariant in Sec. C.1.

6. (**Response to Reviewer 8jGX - 7**) We add a link to visualization of spherical harmonics in Sec. A.3.

---

> ### Comment · Reviewer_8jGX · 2022-11-19
> **Thanks!**
>
> Authors, thank you for your effort to update the manuscript!

---

### Decision · Program_Chairs · 2023-01-20

**Decision:**

Accept: notable-top-25%

**Justification For Why Not Higher Score:**

The paper is not that accessible to non-experts, and conceptually not that innovative.

**Justification For Why Not Lower Score:**

The paper represents significant progress in equivariant network design, and achieves excellent performance on important chemistry problems

**Metareview: Summary, Strengths And Weaknesses:**

This paper proposes a new SE(3)/E(3)-equivariant transformer for atomistic graphs, and demonstrates state of the art performance on well-known benchmark problems. Extensive experiments show that the method outperforms various competing methods, and ablation studies are performed to understand which aspects of the proposed architecture matter most. Reviewers appreciated the architecture and empirical results and insights into architecture design. The main remaining reviewer concerns center around 1) readability, especially for novices 2) lack of clarity about what exactly is novel.

Regarding readability, I agree that the paper could be made more clear. Reviewer 8jGX (currently the only reviewer who does not recommend acceptance) noted that they are not familiar with equivariant networks, and found the paper hard to read. I agree with the judgement that this paper is not really accessible to readers who are not familiar with groups, representations and equivariance, and so this is a weakness. However, in my judgement the equivariant networks community is now large and mature enough that there are enough experts who will still find this paper readable and useful. Furthermore, this paper performs a rather detailed study of equivariant transformer architectures, making it challenging to also fit in a tutorial-style introduction to the basic concepts.

The novelty of the proposed architecture is a bit hard to pinpoint, partly because the authors still do not describe this very clearly in the paper, and partly because the main contribution is indeed "just" a particular way to combine known or somewhat obvious (to experts) network components. However I do not consider this a genuine weakness, for one could use the same argument to disqualify seminal contributions such as the transformer itself. After all, the transformer is just a combination of linear layers, layernorm, skip connections, and softmax nonlinearities. The contribution is in how exactly these components are put together. Similarly, the present paper is mostly an architecture paper that puts together existing (equivariant) building blocks. However, it seems like the authors are not giving themselves enough credit here either, because some components like depth-wise tensor products and equivariant MLP attention seem quite innovative even as layers by themselves. Unlike many other papers in the equivariant networks literature, this paper is not purely focussed on math, but on important engineering choices, which I think is valuable.

In summary, I think this paper is quite important as it introduces a really effective architecture for an important class of problems. Accessibility and clarity wrt contributions is an issue, and I encourage the authors to continue to improve this aspect of the paper, as it could greatly enhance the utility to the community and thus the impact of the paper.

**Note From Pc:**

if the above contains the word "oral" or "spotlight" please see: "oral" presentation means -> notable-top-5% and "spotlight" means -> notable-top-25%. As stated in our emails, we are disassociating presentation type from AC recommendations

**Summary Of Ac-Reviewer Meeting:**

n/a